# FisherSFT: Data-Efficient Supervised Fine-Tuning of Language Models Using Information Gain

**Rohan Deb** [1]  **Kiran Thekumparampil** [2]  **Kousha Kalantari** [2]  **Gaurush Hiranandani** [3]  **Shoham Sabach** [2 4]
**Branislav Kveton** [5]

## Abstract

Supervised fine-tuning (SFT) is the most common way of adapting large language models (LLMs) to a new domain. In this paper, we improve the efficiency of SFT by selecting an informative subset of training examples. Specifically, for a fixed budget of training examples, which determines the computational cost of fine-tuning, we select those that maximize information gain, as measured by the Fisher information matrix of the SFT objective. We approximate it efficiently by linearization at the last layer of the LLM. Our approach is computationally efficient, analyzable, and performs well empirically. We demonstrate this on several problems, with both quantitative results and LLM-as-a-judge evaluations.

## 1. Introduction

*Large language models (LLMs)* (Bommasani et al., 2021) have emerged as general purpose tools that can solve natural language tasks in both zero-shot and few-shot settings (Radford et al., 2019; Brown et al., 2020). LLMs are typically trained in three stages (Ouyang et al., 2022): pre-training on a large corpus of diverse text, supervised fine-tuning in the domain of interest (Wei et al., 2022), and alignment to human preferences (Ouyang et al., 2022; Rafailov et al., 2023). The main challenge in all stages is the sheer scale of LLMs, which increased by four orders of magnitude in just four years: from 117 million parameters in GPT-2 (2019) to 1.76 trillion parameters in GPT-4 (2023).

In this paper, we focus on making *supervised fine-tuning (SFT)* (Wei et al., 2022) more efficient. A standard approach in SFT is to optimize a *low-rank adapter (LoRA)* (Hu et al.,

2022). The key idea in LoRA is to add low-rank matrices to the matrices in the transformer layers and only optimize those during fine-tuning. The computational cost of LoRA and the quality of its approximations increase with the rank of the low-rank matrices. Therefore, the rank is a natural tunable parameter. The simplicity of LoRA made it popular in practice and thousands of different adapters have been trained (Mangrulkar et al., 2022). We propose a complementary approach that selects a subset of most informative training examples for fine-tuning. The computational cost of fine-tuning is linear in the size of the chosen subset. Therefore, as in LoRA, the number of chosen examples naturally trades off the computational cost of fine-tuning for quality.

The idea of selecting better training examples for SFT is not new and has been explored extensively before. Coverage-based approaches (Phillips, 2017; Tukan et al., 2021) select diverse examples to form coresets. Quality-based sampling (Wenzek et al., 2019; Muenchigoff et al., 2023) weeds out low-value or unhelpful examples. In ASK-LLM (Sachdeva et al., 2024), a proxy LLM is prompted with potential training examples and asked if they should be used for training. We review these approaches in detail in Appendix A. The main difference in our work is that we choose training examples using their information value, as measured by the Fisher information matrix of the SFT objective.

Without loss of generality, we view training examples in fine-tuning as sentences, each being a sequence of tokens. We want to select the most informative $n$ sentences, which determines the computational cost of fine-tuning. The key insight in our work is that the SFT objective is a sum of next token log-probabilities, each represented by a multinomial logistic regression model (Bishop, 2006) that depends on all previous tokens. Therefore, the problem of selecting the most informative sentences for fine-tuning is a variant of an optimal design (Pukelsheim, 2006; Stufken & Yang, 2012) for multinomial logistic regression, where the information gain of a sentence depends on all tokens in it. We derive an efficient approximation to the Hessian of the SFT objective, which measures how informative a set of sentences is, and then optimize its lower bound to find the most informative sentences.

---

[1]University of Illinois, Urbana-Champaign. The work was done during an internship at Amazon [2]Amazon [3]Typeface [4]Technion [5]Adobe Research. Correspondence to: Rohan Deb <rd22@illinois.edu>.

*Proceedings of the 42nd International Conference on Machine Learning*, Vancouver, Canada. PMLR 267, 2025. Copyright 2025 by the author(s).

We make the following contributions:

**(1)** We establish a connection between the SFT objective of LLMs and a product of multinomial logistic regression models in Section 2.

**(2)** We propose our method in Section 3. Our main contribution is a computationally-efficient approximation to the log-determinant of the Hessian of the SFT log-likelihood. Specifically, all matrices in this approximation are $d \times d$, where $d$ is the size of transformer embeddings, as opposing to $dL \times dL$, where $L$ is the number of distinct tokens. Since the log-determinant is both monotone and submodular, we maximize it greedily (Nemhauser et al., 1978). We call our algorithm `FisherSFT` because it greedily selects sentences with jointly most informative tokens. This is in a stark contrast to representing sentences by embeddings (Das et al., 2024; Mukherjee et al., 2024; Thekumparampil et al., 2024; Liu et al., 2024; Scheid et al., 2024), which we compare to in Section 5.

**(3)** We analyze `FisherSFT` in Section 4 and show that its prediction error decreases at rate $\tilde{O}(dL/\sqrt{n})$, where $n$ is the number of chosen sentences. The dependence on $n$ is similar to other recent results (Zhu et al., 2023; Mukherjee et al., 2024; Thekumparampil et al., 2024).

**(4)** We evaluate `FisherSFT` empirically in Section 5. Our experiments on synthetic problems show that `FisherSFT` yields a lower prediction error than the baselines. We also fine-tune GPT-2 models and evaluate them using an LLM-as-a-judge (Zheng et al., 2023). The judge prefers the text generated by `FisherSFT` models by a large margin.

## 2. Problem Formulation

*Supervised fine-tuning (SFT)* (Mangrulkar et al., 2022; Hu et al., 2022) is a direct application of supervised learning to LLMs. The objective of SFT is to minimize the negative log-likelihood of LLM responses given prompts. Specifically, let $y_i$ be a response to prompt $z_i$, and $\mathcal{D} = \{(z_i, y_i)\}_{i \in [N]}$ be a dataset $N$ such prompt-response pairs. Then the SFT objective is to minimize

$$-\frac{1}{N} \sum_{i=1}^{N} \log p(y_i \mid z_i), \quad (1)$$

where $p(y_i \mid z_i)$ denotes the probability of $y_i$ given $z_i$ under the LLM. In this work, we select the most informative data points for optimizing (1) by linearizing it, and building on existing results for active learning in linear and generalized linear models (Pukelsheim, 2006; Stufken & Yang, 2012).

The linearization is done as follows. We call $y_i$ a *sentence*. Let sentence $y_i$ consist of $M_i$ tokens indexed by $j \in [M_i]$. Let $y_{i,j} \in [L]$ be the *token* at position $j$ in sentence $i$. The tokens belong to a vocabulary of size $L$. We represent the

sentence $i$ by a sequence of its tokens,

$$y_i = (y_{i,1}, \ldots, y_{i,M_i}).$$

To model the evolution of the sentence token-by-token, we define a vector $\mathbf{x}_{i,j} \in \mathbb{R}^d$ that captures the relevant *history* up to position $j$ in sentence $i$. In large language models, $\mathbf{x}_{i,j}$ is the output of the pre-logit layer that encodes contextual information about tokens $y_{i,1}, \ldots, y_{i,j-1}$ and prompt $z_i$.

Let $\Theta^* = (\theta_\ell^*)_{\ell \in [L]} \in \mathbb{R}^{d \times L}$ be a matrix of parameters in the last logit layer of the LLM and $\theta_\ell^* \in \mathbb{R}^d$ be the parameter vector corresponding to token $\ell \in [L]$. Then the probability of token $\ell$ at position $j$ in sentence $i$ is

$$p(\ell \mid \mathbf{x}_{i,j}; \Theta^*) = \frac{\exp(\theta_\ell^{*\top} \mathbf{x}_{i,j})}{\sum_{k=1}^{L} \exp(\theta_k^{*\top} \mathbf{x}_{i,j})}. \quad (2)$$

Using this notation, the SFT objective in (1) can be equivalently expressed as

$$\mathcal{L}(\Theta) = -\frac{1}{N} \sum_{i=1}^{N} \sum_{j=1}^{M_i} \log p(y_{i,j} \mid \mathbf{x}_{i,j}; \Theta). \quad (3)$$

Our objective is to select a subset of $n$ sentences out of $N$ for fine-tuning. We denote it by $\mathcal{S} \subset [N]$ and assume that $|\mathcal{S}| = n$. The negative log-likelihood on $\mathcal{S}$, which helps us to choose the sentences, is defined as

$$\mathcal{L}_\mathcal{S}(\Theta) = -\frac{1}{n} \sum_{i \in \mathcal{S}} \sum_{j=1}^{M_i} \log p(y_{i,j} \mid \mathbf{x}_{i,j}; \Theta). \quad (4)$$

Let

$$\hat{\Theta} = \underset{\Theta}{\operatorname{argmin}} \, \mathcal{L}_\mathcal{S}(\Theta) \quad (5)$$

be the *maximum likelihood estimate (MLE)* of $\Theta^*$ under the negative log-likelihood in (4).

## 3. Algorithm

The *Hessian* of (4), which is also known as the *Fisher information matrix* (Fisher, 1922), is

$$\nabla^2 \mathcal{L}_\mathcal{S}(\Theta) = -\frac{1}{n} \sum_{i \in S} \sum_{j=1}^{M_i} \nabla^2 \log p(y_{i,j} \mid \mathbf{x}_{i,j}; \Theta). \quad (6)$$

The inverse of this matrix is the covariance matrix of the MLE in (5). Therefore, $\nabla^2 \mathcal{L}_\mathcal{S}(\Theta^*)$ can be used to quantify the uncertainty of $\hat{\Theta}$ and gather training examples for improving our estimate of $\Theta^*$ (Pukelsheim, 2006; Stufken & Yang, 2012; Lattimore & Szepesvari, 2019).

In this paper, we optimize $\nabla^2 \mathcal{L}_\mathcal{S}(\Theta^*)$ by maximizing its eigenvalues with respect to $\mathcal{S}$, and approximate this as maximizing $\log \det(\nabla^2 \mathcal{L}_\mathcal{S}(\Theta^*))$. This optimization problem is

**Algorithm 1** Greedy optimal design for language models.

1: **Input:** Embeddings $\{(\mathbf{x}_{i,j})_{j=1}^{M_i}\}_{i=1}^{N}$
2: Design matrix $V \leftarrow \sigma_0 I_d$
3: Selected sentences $\mathcal{S} \leftarrow \emptyset$
4: **for** $t = 1, \ldots, n$ **do**
5: $\qquad k \leftarrow \underset{i \in [N] \setminus \mathcal{S}}{\operatorname{argmax}} \log \det \left( V + \sum_{j=1}^{M_i} \mathbf{x}_{i,j} \mathbf{x}_{i,j}^{\top} \right)$
6: $\qquad \mathcal{S} \leftarrow \mathcal{S} + \{k\}$
7: $\qquad V \leftarrow V + \sum_{j=1}^{M_k} \mathbf{x}_{k,j} \mathbf{x}_{k,j}^{\top}$
8: **Output:** $\mathcal{S}$

difficult for three reasons. First, $\nabla^2 \mathcal{L}_{\mathcal{S}}(\Theta^*)$ is a $dL \times dL$ matrix. For practical values of $d \approx 10^3$ and $L \approx 10^5$, it is computationally infeasible to optimize $dL \times dL$ matrices. Second, $\nabla^2 \mathcal{L}_{\mathcal{S}}(\Theta^*)$ cannot be optimized directly because $\Theta^*$ is not known. To address these two issues, we derive a lower bound on $\log \det(\nabla^2 \mathcal{L}_{\mathcal{S}}(\Theta))$ that only involves $d \times d$ matrices and does not depend on $\Theta$. We present the lower bound in the following lemma.

**Lemma 3.1.** *Consider the loss function given in* (4) *and let* $p(\,\cdot\,|\mathbf{x}_{i,j}; \Theta)$ *be the probability vector whose $\ell$-th component is* $p(\ell|\mathbf{x}_{i,j}; \Theta)$. *Then the Hessian of the loss is*

$$\nabla^2 \mathcal{L}_{\mathcal{S}}(\Theta) = \frac{1}{n} \sum_{i \in \mathcal{S}} \sum_{j=1}^{M_i} \Big[ \mathbf{diag}(p(\,\cdot\,|\mathbf{x}_{i,j}; \Theta)) \\ - p(\,\cdot\,|\mathbf{x}_{i,j}; \Theta) p(\,\cdot\,|\mathbf{x}_{i,j}; \Theta)^{\top} \Big] \otimes \mathbf{x}_{i,j} \mathbf{x}_{i,j}^{\top},$$

*where $\otimes$ is the tensor product and $\mathbf{diag}(v)$ is a matrix with diagonal $v$. Moreover, if*

$$\mathbf{diag}(p(\,\cdot\,|\mathbf{x}_{i,j}; \Theta)) - p(\,\cdot\,|\mathbf{x}_{i,j}; \Theta) p(\,\cdot\,|\mathbf{x}_{i,j}; \Theta)^{\top} \succeq \gamma$$

*holds for some $\gamma > 0$, then*

$$\log \det(\nabla^2 \mathcal{L}_{\mathcal{S}}(\Theta)) \geq d \log \det \left( \frac{\gamma}{n} \sum_{i \in S} \sum_{j=1}^{M_i} \mathbf{x}_{i,j} \mathbf{x}_{i,j}^{\top} \right).$$

*Proof.* The lemma is proved in Section 3.4. $\qquad\square$

Therefore, instead of maximizing $\log \det(\nabla^2 \mathcal{L}_{\mathcal{S}}(\Theta))$, we can maximize $\log \det(\sum_{i \in \mathcal{S}} \sum_{j=1}^{M_i} \mathbf{x}_{i,j} \mathbf{x}_{i,j}^{\top})$ and this is the key idea in our algorithm `FisherSFT`. The last challenge is that we have a combinatorial optimization problem, choose a subset of $n$ sentences out of $N$. Since $\log \det$ is a monotone submodular function, we solve this problem greedily (Nemhauser et al., 1978).

**Algorithm 2** `FisherSFT`: Fast Implementation of Algorithm 1.

1: **Input:** Embeddings $\{(\mathbf{x}_{i,j})_{j=1}^{M_i}\}_{i=1}^{N}$, batch size $B$
2: Design matrix $V \leftarrow \sigma_0 I_d$
3: Selected sentences $\mathcal{S} \leftarrow \emptyset$
4: Cached information gains $g \leftarrow \infty_N$
5: **for** $t = 1, \ldots, n$ **do**
6: $\qquad g_{\max} \leftarrow 0$
7: $\qquad$ **for** $b = 1, \ldots, N/B$ **do**
8: $\qquad\qquad \mathcal{B} \leftarrow \{(b-1)B + 1, \ldots, bB\}$
9: $\qquad\qquad$ **for all** $i \in \mathcal{B}$ **do**
10: $\qquad\qquad\qquad$ **if** $g_i > g_{\max}$ **then**
11: $\qquad\qquad\qquad\qquad g_i \leftarrow \log \det \left( V + \sum_{j=1}^{M_i} \mathbf{x}_{i,j} \mathbf{x}_{i,j}^{\top} \right) - \\ \qquad\qquad\qquad\qquad\qquad \log \det(V)$
12: $\qquad\qquad g_{\max} \leftarrow \max \left( \{g_i\}_{i \in \mathcal{B}} + \{g_{\max}\} \right)$
13: $\qquad k \leftarrow \operatorname{argmax}_{i \in [N] \setminus \mathcal{S}} g_i$
14: $\qquad \mathcal{S} \leftarrow \mathcal{S} + \{k\}$
15: $\qquad V \leftarrow V + \sum_{j=1}^{M_k} \mathbf{x}_{k,j} \mathbf{x}_{k,j}^{\top}$
16: **Output:** $\mathcal{S}$

### 3.1. Greedy Optimal Design

Our algorithm is presented in Algorithm 1 and we explain it next. We call the Hessian of the SFT objective a *design matrix* because we use it to design the set of chosen sentences. The design matrix is initialized as $V = \sigma_0 I_d$ (line 2), where $\sigma_0 \geq 0$ is the strength of regularization. The set of chosen sentences is initialized as $\mathcal{S} = \emptyset$ (line 3). In step $t \in [n]$, we select a sentence from the remaining sentences $[N] \setminus \mathcal{S}$ that maximizes $\log \det$ of the design matrix of the previously chosen sentences (line 5). This sentence has the highest information gain. Intuitively, it contains the most diverse embeddings $\mathbf{x}_{i,j}$ since $\log \det(V)$ is the logarithm of the volume of the ellipsoid represented by $V$, and this is maximized when the lengths of all its axes increase equally (Lattimore & Szepesvari, 2019). After the sentence is chosen, we add it to the current subset of sentences $\mathcal{S}$ (line 6) and $\sum_{j=1}^{M_k} \mathbf{x}_{k,j} \mathbf{x}_{k,j}^{\top}$ is added to the design matrix $V$ (line 7).

Algorithm 1 selects one sentence per step (line 5). In each step, we compute $\log \det$ for all remaining sentences. This is clearly not practical. In Section 3.2, we present a faster algorithm that leverages the submodularity of $\log \det$ and parallelism to produce the same set of sentences $\mathcal{S}$.

### 3.2. Algorithm `FisherSFT`

Now we present a more computationally-efficient variant of Algorithm 1 that exploits the submodularity of $\log \det$ and parallelism (Algorithm 2). Simply put, we implement line 5

in Algorithm 1 more efficiently. This corresponds to line 13 in Algorithm 2.

The key idea is to cache information gains, where $g_i$ is the cached information gain for sentence $i \in [N]$. The gains are initialized as $g_i = \infty$ (line 4), updated in line 11, and we act greedily with respect to them in line 13. If the gains were always updated, line 13 would be clearly identical to line 5 in Algorithm 1 because the maximized values in line 13 are only offset by a constant $\log \det(V)$.

The key insight to efficient updates is that $\log \det$ is a monotone submodular function. Therefore, the gains cannot increase as $V$ is updated and hence do not have to be recomputed when they are smaller than the highest gain $g_{\max}$ at step $t \in [n]$ thus far. We exploit this structure in line 10 and update $g_{\max}$ in line 12. Finally, we update $g_i$ in batches of size $B$ (line 9) in parallel. This results in an additional $O(B)$ speedup. We use this implementation in our experiments and call it `FisherSFT`.

### 3.3. Discussion

We consider two variants of `FisherSFT` in this work. In Section 4, we analyze an idealized variant where the pre-logit layer of the LLM is treated as a fixed feature vector. After $n$ sentences are chosen by `FisherSFT`, we estimate the model parameter $\Theta^*$ by maximizing the log-likelihood in (5). We argue that $\hat{\Theta}$ approaches $\Theta^*$ as the sample size $n$ increases in Section 4. We experiment with this setting in Sections 5.1 and 5.2.

We fine-tune LLMs in Section 5.3. The main difference in this setting is that $\mathcal{L}_{\mathcal{S}}(\Theta)$ is only used to select a subset of $n$ sentences but we fine-tune the LLM instead of solving (5). The structure of the LLM is taken into account using embeddings $\mathbf{x}_{i,j}$. At a high level, `FisherSFT` chooses diverse sentences to ensure a more uniform coverage of $\mathbf{x}_{i,j}$. This reduces the original $dL \times dL$ Fisher information matrix optimization into a tractable $d \times d$ problem, enabling efficient optimization. We discuss several notable aspects of our approximation next.

The Fisher information matrix in `FisherSFT` is derived in Lemma 3.1. The algebraic form of this matrix, the outer product of feature vectors, would be the true Fisher information matrix in linear models. Optimization of this matrix leads to choosing feature vectors that uniformly cover all directions spanned by $\mathbf{x}_{i,j}$. Therefore, `FisherSFT` controls the worst-case prediction error over $\mathbf{x}_{i,j}$ and hence ensures robustness. We observe this in our experiments. This provides a different perspective than Sorscher et al. (2022) who show that it is possible to break beyond power law scaling by intelligently choosing easy and hard training examples.

`FisherSFT` is naturally biased towards selecting longer sentences. This bias arises because each token in a sentence

is a training point in SFT, for the conditional probability of token $y_{i,j}$ given the history embedding $\mathbf{x}_{i,j}$. Therefore, longer sentences may have higher information gains. Nevertheless, note that the sentences in `FisherSFT` are chosen based on their total information gain and not just length.

The key idea in our work can be viewed as using the last-layer embedding in LLMs as a featurizer (Xu et al., 2022). This is known to yield robust performance in active learning (Riquelme et al., 2018). Other popular active learning techniques utilize gradients (Ash et al., 2020) or are based on bandit exploration (Zhou et al., 2020; Zhang et al., 2021). These methods are computationally prohibitive in modern neural networks because of the sheer size of their parameter spaces.

### 3.4. Proof of Lemma 3.1

In Proposition B.1, we show that

$$\nabla^2 \mathcal{L}_{\mathcal{S}}(\Theta) = \frac{1}{n} \sum_{i \in \mathcal{S}} \sum_{j=1}^{M_i} \Big( \mathbf{diag}(p(\,\cdot\,|\mathbf{x}_{i,j}; \Theta)) \\ - p(\,\cdot\,|\mathbf{x}_{i,j}; \Theta) p(\,\cdot\,|\mathbf{x}_{i,j}; \Theta)^\top \Big) \otimes \mathbf{x}_{i,j} \mathbf{x}_{i,j}^\top .$$

Now suppose that

$$\mathbf{diag}(p(\,\cdot\,|\mathbf{x}_{i,j}; \Theta)) - p(\,\cdot\,|\mathbf{x}_{i,j}; \Theta) p(\,\cdot\,|\mathbf{x}_{i,j}; \Theta)^\top \succeq \gamma$$

holds for some $\gamma > 0$. Then using Theorem 4.2.12 of Horn & Johnson (1991), we get

$$\nabla^2 \mathcal{L}_{\mathcal{S}}(\Theta) \succeq \frac{1}{n} \sum_{i \in \mathcal{S}} \sum_{j=1}^{M_i} \gamma I_L \otimes \mathbf{x}_{i,j} \mathbf{x}_{i,j}^\top ,$$

where $I_L$ is an $L$-dimensional identity matrix. In turn,

$$\det(\nabla^2 \mathcal{L}_{\mathcal{S}}(\Theta)) \geq \det\Big( I_L \otimes \frac{\gamma}{n} \sum_{i \in S} \sum_{j=1}^{M_i} \mathbf{x}_{i,j} \mathbf{x}_{i,j}^\top \Big) .$$

Now using the fact that $\det(A \otimes B) = \det(A)^p \det(B)^q$ for $A \in \mathbb{R}^{p \times p}$ and $B \in \mathbb{R}^{q \times q}$ (Bernstein, 2009, Proposition 7.1.11), we have

$$\det(\nabla^2 \mathcal{L}_{\mathcal{S}}(\Theta)) \geq \det(I_L)^L \det\Big( \frac{\gamma}{n} \sum_{i \in S} \sum_{j=1}^{M_i} \mathbf{x}_{i,j} \mathbf{x}_{i,j}^\top \Big)^d$$

$$= \det\Big( \frac{\gamma}{n} \sum_{i \in S} \sum_{j=1}^{M_i} \mathbf{x}_{i,j} \mathbf{x}_{i,j}^\top \Big)^d .$$

Taking the logarithm of both sides completes the proof.

## 4. Error Bound

Our main Theorem 4.3 provides a $O(1/\sqrt{n})$ bound on the *maximum prediction error* of the estimated parameter $\hat{\Theta}$

from the sentences collected by `FisherSFT`. The *maximum prediction error* is given by

$$\max_{i\in[N]}\sum_{j=1}^{M_i}\|\Theta^{*\top}\mathbf{x}_{i,j}-\hat{\Theta}^\top\mathbf{x}_{i,j}\|_2\,.$$

Note that $\|\cdot\|_2$ is the sum of prediction errors over a vocabulary of size $L$. We make the following assumption on the feature vectors and unknown parameter $\Theta^*$.

**Assumption 4.1.** Let $\|\mathbf{x}_{i,j}\|\le 1$ for any $i\in[N]$ and $j\in[M_i]$. Moreover, we assume that the true model parameter satisfies $\Theta^*\in\mathcal{B}$, where

$$\mathcal{B}=\Big\{\Theta=(\theta_\ell)_{\ell\in[L]}:\theta_\ell\in\mathbb{R}^d,\|\theta_\ell\|_2\le 1,$$

$$\sum_{k=1}^d\theta_{\ell,k}=0,\forall\ell\in[L]\Big\}\,.$$

For any subset $\mathcal{S}\subseteq[N]$, we define the regularized design matrix for $\mathcal{S}$ as

$$\bar{\Sigma}_\mathcal{S}=\sigma_0 I_d+\sum_{i\in\mathcal{S}}\sum_{j=1}^{M_i}\mathbf{x}_{i,j}\mathbf{x}_{i,j}^\top\qquad(7)$$

for $\sigma_0\ge 0$. `FisherSFT` iteratively computes this matrix as new sentences are added to the subset $\mathcal{S}$ (line 15).

Next we make a diversity assumption on our dataset.

**Assumption 4.2.** Let $\mathcal{S}_{t-1}$ be the set of chosen sentences up to step $t$ in `FisherSFT` and $I_t$ be the index of the chosen sentence at step $t$. We assume that there exists a constant $\kappa\ge 1$ such that for any step $t\in[n]$ and sentence $i\in[N]$,

$$\log\det(I_d+\sum_{j=1}^{M_i}\bar{\Sigma}_{t-1}^{-1/2}\mathbf{x}_{i,j}\mathbf{x}_{i,j}^\top\bar{\Sigma}_{t-1}^{-1/2})$$

$$\le\kappa\log\det(I_d+\sum_{j=1}^{M_{I_t}}\bar{\Sigma}_{t-1}^{-1/2}\mathbf{x}_{I_t,j}\mathbf{x}_{I_t,j}^\top\bar{\Sigma}_{t-1}^{-1/2})\,,$$

where $\bar{\Sigma}_{t-1}=\bar{\Sigma}_{\mathcal{S}_{t-1}}$.

The assumption says that the information gain of the chosen sentence (line 13 of `FisherSFT`) is an approximate upper bound, up to a multiplicative $\kappa\ge 1$, on the information gain of any other sentence, including those previously chosen that cannot be chosen again.

Now we state our main result.

**Theorem 4.3.** *Suppose that Assumptions 4.1 and 4.2 hold. Then for any $\delta>0$, the maximum prediction error of $\hat{\Theta}$ is bounded with probability $1-\delta$ as*

$$\max_{i\in[N]}\sum_{j=1}^{M_i}\|\Theta^{*\top}\mathbf{x}_{i,j}-\hat{\Theta}^\top\mathbf{x}_{i,j}\|_2$$

$$\le CMe^2L\sqrt{\frac{\sigma_0^{-2}\log\left(1+\frac{\sigma_0^{-2}nM}{d}\right)}{\log(1+\sigma_0^{-2})}}\sqrt{\frac{d\kappa(d+\log(L/\delta))}{n}}\,,$$

*where $C>0$ is some global constant.*

We sketch the proof next. Let $\mathcal{S}$ be the set of $n$ sentences generated by `FisherSFT`. For any positive-definite matrix $A\in\mathbb{R}^{d\times d}$ and vector $x\in\mathbb{R}^d$, we define $\|x\|_A=\sqrt{x^\top Ax}$. With $\hat{\Theta}=(\hat{\theta}_\ell)_{\ell\in[L]}$ and $\Theta^*=(\theta_\ell^*)_{\ell\in[L]}$, we can bound the maximum prediction error as

$$\max_{i\in[N]}\sum_{j=1}^{M_i}\|\hat{\Theta}^\top\mathbf{x}_{i,j}-\Theta^{*\top}\mathbf{x}_{i,j}\|_2$$

$$\le\max_{i\in[N]}\sum_{j=1}^{M_i}\sum_{\ell=1}^L|(\hat{\theta}_\ell-\theta_\ell^*)^\top\mathbf{x}_{i,j}|$$

$$\le\max_{i\in[N]}\sum_{j=1}^{M_i}\sum_{\ell=1}^L\|\hat{\theta}_\ell-\theta_\ell^*\|_{\bar{\Sigma}_\mathcal{S}}\|\mathbf{x}_{i,j}\|_{\bar{\Sigma}_\mathcal{S}^{-1}}$$

$$\le\underbrace{\Big(\sum_{\ell=1}^L\|\hat{\theta}_\ell-\theta_\ell^*\|_{\bar{\Sigma}_\mathcal{S}}\Big)}_{\text{I}}\underbrace{\max_{i\in[N]}\sum_{j=1}^{M_i}\|\mathbf{x}_{i,j}\|_{\bar{\Sigma}_\mathcal{S}^{-1}}}_{\text{II}}\,.\quad(8)$$

Term I represents a self-normalized error between the true parameter $\Theta^*$ and its MLE $\hat{\Theta}$. Term II measures the information value of sentences $\mathcal{S}$. We start with bounding term II. Under Assumption 4.2, we get the following bound.

**Lemma 4.4.** *Suppose that Assumption 4.2 holds. Let $\mathcal{S}$ be the set of $n$ sentences generated by `FisherSFT` and (7) be their design matrix. Let $M=\max_{i\in[N]}M_i$. Then*

$$\max_{i\in[N]}\sum_{j=1}^{M_i}\|\mathbf{x}_{i,j}\|_{\bar{\Sigma}_\mathcal{S}^{-1}}^2\le\frac{\sigma_0^{-2}\log\left(1+\frac{\sigma_0^{-2}nM}{d}\right)}{\log(1+\sigma_0^{-2})}\frac{\kappa dM}{n}\,.$$

*Proof of Lemma 4.4.* See Appendix C.1 □

Using the Cauchy-Schwarz inequality, we get a corollary

$$\max_{i\in[N]}\sum_{j=1}^{M_i}\|\mathbf{x}_{i,j}\|_{\bar{\Sigma}_\mathcal{S}^{-1}}\le M\sqrt{\frac{\sigma_0^{-2}\log\left(1+\frac{\sigma_0^{-2}nM}{d}\right)}{\log(1+\sigma_0^{-2})}\frac{\kappa d}{n}}\,.$$

We bound term I in (8) by relating it to the difference between the loss and its first order approximation,

$$\mathcal{L}_\mathcal{S}(\hat{\Theta})-\mathcal{L}_\mathcal{S}(\Theta^*)-\langle\nabla\mathcal{L}_\mathcal{S}(\Theta^*),\hat{\Theta}-\Theta^*\rangle$$

$$\overset{(a)}{\le}-\langle\nabla\mathcal{L}_\mathcal{S}(\Theta^*),\hat{\Theta}-\Theta^*\rangle$$

$$=-\sum_{\ell=1}^L\nabla_\ell\mathcal{L}_\mathcal{S}(\Theta^*)^\top(\hat{\theta}_\ell-\theta_\ell^*)$$

$$\overset{(b)}{\le}\sum_{\ell=1}^L\|\nabla_\ell\mathcal{L}_\mathcal{S}(\Theta^*)\|_{\bar{\Sigma}_\mathcal{S}^{-1}}\|\hat{\theta}_\ell-\theta_\ell^*\|_{\bar{\Sigma}_\mathcal{S}}\,,\quad(9)$$

where the dot product between matrices $A$ and $B$ is defined as $\langle A, B \rangle = \sum_{i,j} A_{i,j} B_{i,j}$. Inequality $(a)$ is a consequence of $\mathcal{L}_{\mathcal{S}}(\hat{\Theta}) \leq \mathcal{L}_{\mathcal{S}}(\Theta^*)$, because $\hat{\Theta}$ is the minimizer in (5). Inequality $(b)$ follows from the Cauchy-Schwarz inequality. Finally, we bound the left-hand side of (9) from below using the fact that the loss is strongly convex.

**Lemma 4.5.** *Suppose that Assumption 4.1 holds and $\hat{\Theta}$ is the MLE in (5) such that $\hat{\Theta} \in \mathcal{B}$. Then there exists some $\alpha < 1$ such that*

$$\mathcal{L}_{\mathcal{S}}(\hat{\Theta}) - \mathcal{L}_{\mathcal{S}}(\Theta^*) - \langle \nabla \mathcal{L}_{\mathcal{S}}(\Theta^*), \hat{\Theta} - \Theta^* \rangle$$
$$\geq \frac{e^{-2\alpha}}{L} \left( \sum_{\ell=1}^{L} \|\hat{\theta}_\ell - \theta_\ell^*\|_{\bar{\Sigma}_{\mathcal{S}}} \right)^2 .$$

*Proof of Lemma 4.5.* See Appendix C.2 □

Using Lemma 4.5 and (9), we have

$$\frac{e^{-2\alpha}}{L} \left( \sum_{\ell=1}^{L} \|\hat{\theta}_\ell - \theta_\ell^*\|_{\bar{\Sigma}_{\mathcal{S}}} \right)^2$$
$$\leq \sum_{\ell=1}^{L} \left\| \nabla_\ell \mathcal{L}_{\mathcal{S}}(\Theta^*) \right\|_{\bar{\Sigma}_{\mathcal{S}}^{-1}} \|\hat{\theta}_\ell - \theta_\ell^*\|_{\bar{\Sigma}_{\mathcal{S}}}$$
$$\leq \left( \sup_{\ell \in [L]} \left\| \nabla_\ell \mathcal{L}_{\mathcal{S}}(\Theta^*) \right\|_{\bar{\Sigma}_{\mathcal{S}}^{-1}} \right) \left( \sum_{\ell=1}^{L} \|\hat{\theta}_\ell - \theta_\ell^*\|_{\bar{\Sigma}_{\mathcal{S}}} \right) ,$$

and therefore

$$\sum_{\ell=1}^{L} \|\hat{\theta}_\ell - \theta_\ell^*\|_{\bar{\Sigma}_{\mathcal{S}}} \leq e^{2\alpha} L \sup_{\ell \in [L]} \left\| \nabla_\ell \mathcal{L}_{\mathcal{S}}(\Theta^*) \right\|_{\bar{\Sigma}_{\mathcal{S}}^{-1}}$$
$$\leq e^2 L \sup_{\ell \in [L]} \left\| \nabla_\ell \mathcal{L}_{\mathcal{S}}(\Theta^*) \right\|_{\bar{\Sigma}_{\mathcal{S}}^{-1}} . \quad (10)$$

The next lemma bounds $\sup_{\ell \in [L]} \left\| \nabla_\ell \mathcal{L}_{\mathcal{S}}(\Theta^*) \right\|_{\bar{\Sigma}_{\mathcal{S}}^{-1}}$.

**Lemma 4.6.** *With probability $1 - \delta$, the gradient of the loss satisfies*

$$\sup_{\ell \in [L]} \left\| \nabla_\ell \mathcal{L}_{\mathcal{S}}(\Theta^*) \right\|_{\bar{\Sigma}_{\mathcal{S}}^{-1}} \leq C \sqrt{d + \log(L/\delta)} ,$$

*where $C > 0$ is some global constant.*

*Proof of Lemma 4.6.* See Appendix C.3 □

Now we combine (8), the corollary of Lemma 4.4, (10), and Lemma 4.6, and get that

$$\max_{i \in [N]} \sum_{j=1}^{M_i} \|\Theta^{*\top} \mathbf{x}_{i,j} - \hat{\Theta}^\top \mathbf{x}_{i,j}\|_2$$
$$\leq C M e^2 L \sqrt{\frac{\sigma_0^{-2} \log\left(1 + \frac{\sigma_0^2 n M}{d}\right)}{\log(1 + \sigma_0^2)}} \sqrt{\frac{d\kappa(d + \log(L/\delta))}{n}}$$

holds with probability $1 - \delta$, where $C > 0$ is some constant. This completes the proof.

## 5. Experiments

We empirically evaluate FisherSFT on a variety of problems. We experiment with a synthetic autoregressive prediction task in Section 5.1, with pre-trained word embeddings in Section 5.2, and with GPT-2 models in Section 5.3. Ablation studies are conducted in Section 5.4. Our implementation is available at github.

### 5.1. Synthetic Experiments

We start with a simplified setup where each token $\ell \in [L]$ is associated with a vector sampled from a standard normal distribution, $\mathbf{x}_\ell \sim \mathcal{N}(\mathbf{0}, I_d)$. The number of tokens is $L = 20$ and $d = 10$. All entries of $\Theta_*$ are sampled i.i.d. from $\mathcal{N}(0, 1)$. The first token in sentences is sampled uniformly at random from $[L]$. The next tokens are sampled from the softmax model in (2), where $\mathbf{x}_{i,j}$ is the token embedding at position $j - 1$ in sentence $i$.

We consider several baselines. Uniform selects sentences uniformly at random. SentenceOD selects sentences greedily by maximizing the log-determinant of a sentence-level Fisher information matrix. We construct sentence embeddings by summing all token embeddings in that sentence. Specifically, $\mathbf{x}_i = \sum_{j=1}^{M_i} \mathbf{x}_{i,j}$ is the embedding of sentence $i \in [N]$. DensitySampling (Sachdeva et al., 2024) selects sentences based on inverse propensity scores estimated by a kernel density estimator. ClusteredSampling (Axiotis et al., 2024) clusters sentence embeddings using $k$-means and then samples them proportionally to their distance to the closest mean plus its loss. The baselines are described in detail in Appendix A.

All methods choose $n$ sentences and learn a multinomial logistic regression model by solving (5). We evaluate the methods by two metrics: *maximum prediction error*

$$\mathcal{E}_{\max}(n) = \max_{i \in [N]} \sum_{j=1}^{M_i} \|\Theta^{*\top} \mathbf{x}_{i,j} - \hat{\Theta}^\top \mathbf{x}_{i,j}\|_2$$

and *mean prediction error*

$$\mathcal{E}_{mean}(n) = \frac{1}{N} \sum_{i \in [N]} \sum_{j=1}^{M_i} \|\Theta^{*\top} \mathbf{x}_{i,j} - \hat{\Theta}^\top \mathbf{x}_{i,j}\|_2 .$$

The maximum error measures the performance on the most challenging sentence, while the mean error measures the average performance on all sentences. We bound the maximum prediction error of FisherSFT in Theorem 4.3.

The prediction errors of all methods are reported in Figure 1. We observe that FisherSFT performs better than all baselines in both metrics. In fact, it is much more sample efficient than the best baseline. As an example, the lowest maximum prediction error of the best baseline, attained at $n = 2\,000$, is attained by FisherSFT at $n = 1\,000$.

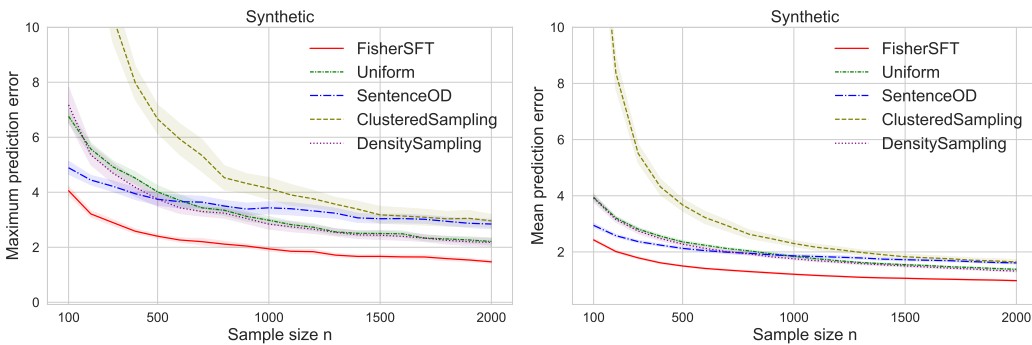

*Figure 1.* Comparison of maximum and mean prediction errors on synthetic tokens. The x axis shows the number of selected sentences for training the model. The y axis shows the errors averaged over 20 runs.

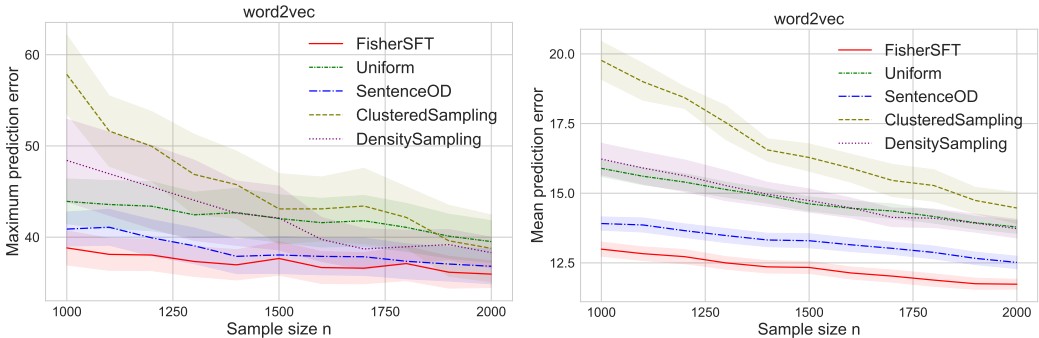

*Figure 2.* Comparison of maximum and mean prediction errors on word2vec tokens. The x axis shows the number of selected sentences for training the model. The y axis shows the errors averaged over 20 runs.

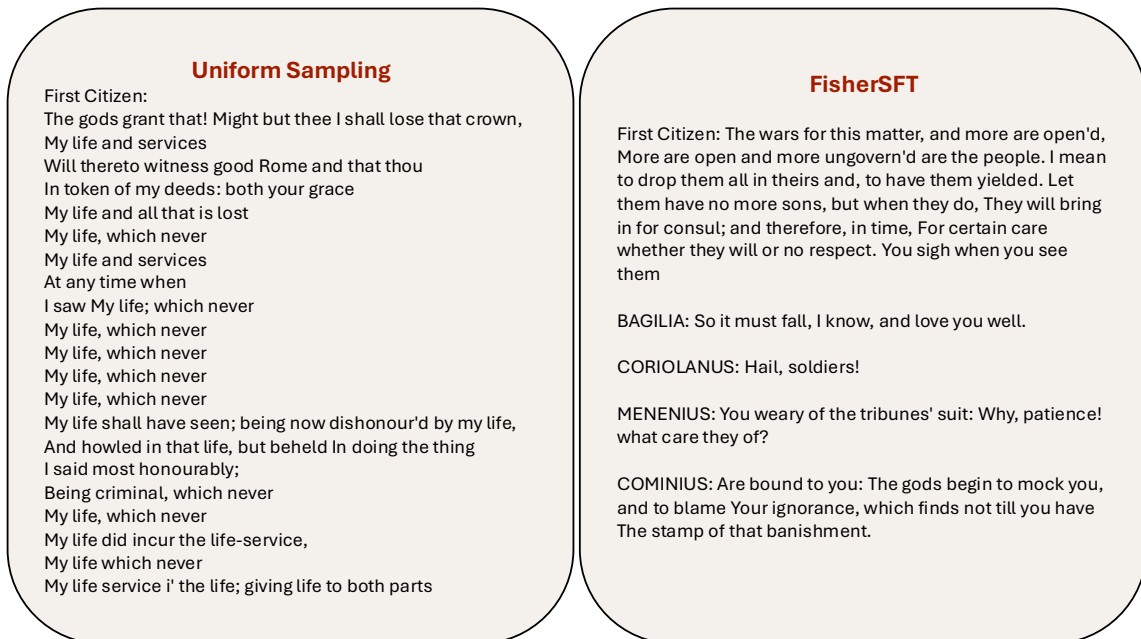

*Figure 3.* Text generated by fine-tuned GPT-2 models on sentences chosen by Uniform and FisherSFT. The latter is more coherent.

## 5.2. Word2vec Embeddings

This experiment is similar to Section 5.1. The difference is that we use pre-trained word2vec embeddings (Mikolov et al., 2013) of dimension 300. We randomly select $L = 20$ words from the word2vec vocabulary and project their embeddings randomly to $d = 10$ dimensions. The vector associated with token $\ell \in [L]$ is $\mathbf{x}_\ell$. The prediction errors of all methods are reported in Figure 2. We observe again that FisherSFT outperforms all baselines in both metrics. As an example, the lowest mean prediction error of the best baseline, attained at $n = 2\,000$, is attained by FisherSFT at $n = 1\,250$.

## 5.3. Experiments with GPT-2

We experiment with GPT-2 models (Radford et al., 2019) next. All methods select a subset of $n$ sentences and then fine-tune the GPT-2 model on Hugging Face (Wolf et al., 2020). The vector $\mathbf{x}_{i,j}$ is the output of the pre-logit layer at position $j$ in sentence $i$ and $y_{i,j}$ is the token at that position. We consider two corpora: tiny Shakespeare corpus (Karpathy, 2015) and Sherlock Holmes corpus (Doyle). We subsample $10\,000$ sentences from both corpora and experiment with learning from $n \in [100, 5\,000]$ sentences.

We choose two baselines from the previous experiments: Uniform and DensitySampling. DensitySampling is chosen because it tends to outperform other baselines on language model fine-tuning tasks (Sachdeva et al., 2024). We also experiment with AskLLM (Sachdeva et al., 2024), where a proxy LLM is prompted with potential training examples and asked if they should be used for training. The baselines are described in detail in Appendix A.

Unlike in Sections 5.1 and 5.2, the true model parameter is unknown, and hence the prediction errors cannot be computed. To address this, we evaluate the fine-tuned models based on their generated text. Specifically, we prompt them with a few words from the original text corpus (represented by tokens $y_{i,1}, \ldots, y_{i,p}$) and the model completes the sentence by generating $y_{i,p+1}, y_{i,p+2}, \ldots, y_{i,1024}$. We evaluate the quality of the completed sentences using a larger GPT-4o model, which serves as an LLM-as-a-judge (Zheng et al., 2023). For the Shakespeare corpus, we use prompt

```
You are a judge of Shakespeare text.
<tag1>text1</tag1>
<tag2>text2</tag2>
Respond 2 if the text inside <tag2>
is more fluent Shakespeare text
than the text inside <tag1>.
Respond 1 otherwise.
```

For the Sherlock Holmes corpus, we replace "Shakespeare" with "Sherlock".

The prompt does not name the compared methods and tar-

| Method | Number of selected sentences | | | | | |
|---|---|---|---|---|---|---|
| FisherSFT vs | 100 | 200 | 500 | 1000 | 2000 | 5000 |
| Uniform | 0.80 | 0.56 | 0.60 | 0.59 | 0.64 | 0.74 |
| DensitySampling | 0.61 | 0.66 | 0.68 | 0.62 | 0.54 | 0.84 |
| AskLLM | 0.59 | 0.52 | 0.68 | 0.59 | 0.68 | 0.74 |

*Table 1.* Win rates of FisherSFT with respect to three baselines on the Shakespeare corpus. A higher win rate than $0.5$ means that FisherSFT is preferred.

| Method | Number of selected sentences | | | | | |
|---|---|---|---|---|---|---|
| FisherSFT vs | 100 | 200 | 500 | 1000 | 2000 | 5000 |
| Uniform | 0.84 | 0.81 | 0.75 | 0.64 | 0.65 | 0.93 |
| DensitySampling | 0.68 | 0.75 | 0.69 | 0.74 | 0.65 | 0.88 |
| AskLLM | 0.74 | 0.58 | 0.65 | 0.60 | 0.61 | 0.89 |

*Table 2.* Win rates of FisherSFT with respect to three baselines on the Sherlock Holmes corpus. A higher win rate than $0.5$ means that FisherSFT is preferred.

gets the expected benefit of our method, that it yields more fluent, coherent, and natural text. The text generated by the compared methods is randomized: one randomly-chosen method replaces text1 and the other text2. We use the same initial phrase to generate text1 and text2. The LLM judge chooses the first position with probability $0.54$, which is sufficiently close to a completely unbiased $0.5$. Examples of text1 and text 2 are shown in Figure 3. The model trained on uniformly selected sentences generates worse text, which is repetitive. In contrast, the FisherSFT text is more coherent and similar to the Shakespeare corpus.

We evaluate FisherSFT by its win rate, the fraction of time that its responses are judged as more fluent than the baseline. The win rate is estimated from 100 runs. A higher win rate than $0.5$ means that FisherSFT outperforms the baseline. We report the results on Shakespeare and Sherlock Holmes corpora in Tables 1 and 2, respectively. Note that all win rates are higher than $0.5$. Notably, all win rates except one in Table 2 are at least $0.6$. Therefore, FisherSFT performs better than the baselines by a large margin.

## 5.4. Ablation Studies

**Convergence rate.** We validate the empirical convergence rate of FisherSFT in Figure 4a. Specifically, we take the synthetic problem in Section 5.1 and plot the logarithm of the error rate as a function of the logarithm of the sample size. We observe a slope of $-0.3$, which validates that the error rate is $O(n^p)$. We believe that this is sufficiently close to the expected $p = -0.5$, especially since other factors may have played a role at our small sample sizes.

**Computation time.** We report the computation times of the slow and fast implementations of FisherSFT in Algo-

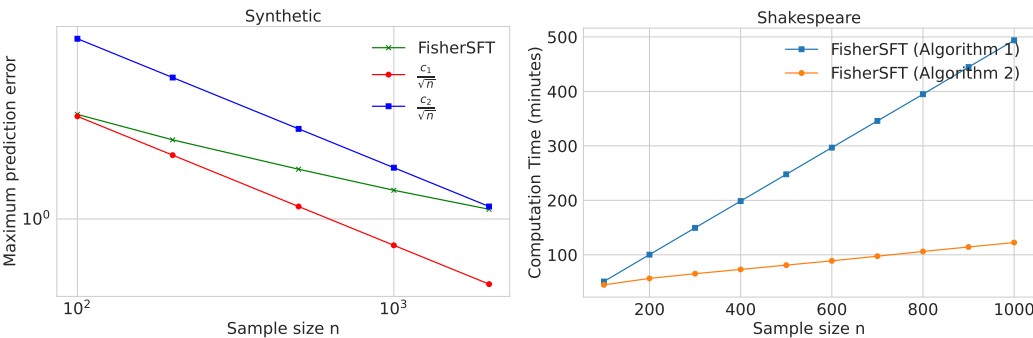

*Figure 4.* a: A log-log plot of the error rate of `FisherSFT` as a function of the sample size $n$. b: Computation times of the slow and fast implementations of `FisherSFT` in Algorithms 1 and 2, respectively.

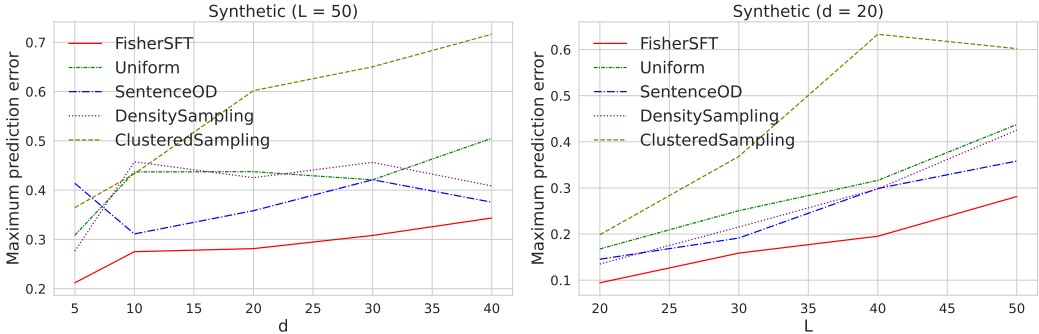

*Figure 5.* Error rates of `FisherSFT` and all baselines from Section 5.1 as functions of the embedding dimension $d$ and vocabulary size $L$.

rithms 1 and 2, respectively, in Figure 4b. The experiment is done on $5\,000$ sentences from the Shakespeare dataset in Section 5.3. We plot the computation times for various sample sizes $n$ and observe that the fast implementation is about 4 times faster.

**Trends in $d$ and $L$.** We plot the error rates of `FisherSFT` and all baselines from Section 5.1 as functions of the embedding dimension $d$ and vocabulary size $L$ in Figure 5. We observe that `FisherSFT` has the lowest error in all settings, indicating robustness to changes in the embedding dimension and vocabulary size.

## 6. Conclusions

We improve the efficiency of supervised fine-tuning by selecting the most informative training examples. The examples maximize a novel approximation to the information gain of the SFT objective. The approximation is computationally efficient and motivates our algorithm `FisherSFT`. We analyze `FisherSFT`, and evaluate it empirically on both synthetic problems and fine-tuning GPT-2 models. We observe that `FisherSFT` has a lower prediction error than the baselines and its fine-tuned models are preferred over the baselines in our LLM-as-a-judge evaluations.

Our work can be extended in several directions. First, we only experiment with a small GPT-2 model. Second, our fine-tuned models are evaluated by an LLM-as-a-judge but not humans. Finally, note that the optimal design could be computed using a different embedding than that from the fine-tuned LLM. This could lead to computational savings if the embedding had a much lower dimensionality.

## Impact Statement

This paper presents work whose goal is to advance the field of Machine Learning. There are many potential societal consequences of our work, none which we feel must be specifically highlighted here.

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

## A. Related Works

---

**Algorithm 3** Inverse Propensity Sampling (IPS) via Kernel Density Estimation (KDE) (Sachdeva et al., 2021)

1: Dataset $\mathcal{D} = \{x_1, x_2, \ldots, x_N\}$ of embeddings, sample size $k$, kernel $k$ with corresponding LSH family $\mathcal{H}$ (Coleman & Shrivastava, 2020), hash range $B$, rows $R$, random seed $s$.
2: Ensure a subset of $\mathcal{D}$ of size $k$, sampled with probability $p$ (see line 14).
3: Initialize KDE sketch $S \leftarrow \mathbf{0}^{R \times B}$.
4: Generate $R$ independent hash functions $h_1, \ldots, h_R$ from $\mathcal{H}$ with range $B$ and random seed $s$.
5: **for** $n \leftarrow 1$ to $N$ **do**
6:     **for** $r \leftarrow 1$ to $R$ **do**
7:         $S_{r, h_r(x_n)} \leftarrow S_{r, h_r(x_n)} + 1$
8: Initialize a list of scores $\mathcal{S} \leftarrow [\,]$.
9: **for** $n \leftarrow 1$ to $N$ **do**
10:     $score \leftarrow 0$
11:     **for** $r \leftarrow 1$ to $R$ **do**
12:         $score \leftarrow score + S[r, h_r(x_n)]$
13:     Append $\frac{score}{R}$ to $\mathcal{S}$.
14: **Output:** Select $k$ elements from $\mathcal{D}$ with probability $p = \frac{S}{\sum S}$ (sampled without replacement).

---

Coverage-oriented approaches center on ensuring that a training set reflects the entire input distribution as broadly as possible. One common strategy is *cluster sampling* (Lee et al., 2023), which embeds data points in a metric space (often via learned representations) and selects mutually distant examples to form "coresets" (Phillips, 2017; Tukan et al., 2021). Related methods include *prototype-based sampling* for vision (Sorscher et al., 2022) and *deduplication algorithms* (Abbas et al., 2023; Lee et al., 2022; Tirmala et al., 2023) that remove near-duplicates or redundancies. More sophisticated procedures—such as *submodular optimization* (Chen et al., 2012; Indyk et al., 2014; Borsos et al., 2020) and *discrepancy minimization* (Karnin & Liberty, 2019)—further refine coverage by balancing representation across diverse data regions.

Quality-based sampling, in contrast, prioritizes weeding out low-value or unhelpful examples. A prominent technique is *perplexity filtering* (Wenzek et al., 2019; Muenchigoff et al., 2023), which prefers samples with higher likelihood under a pretrained model, though this can inadvertently discard valuable but rare text. Other approaches compute "uncertainty scores" via ensemble disagreement (Chitta et al., 2021; Meding et al., 2021) or examine whether examples are *memorized* (Feldman & Zhang, 2020) or *unlearnable* (Mindermann et al., 2022). The *SVP algorithm* (Coleman et al., 2020; Sachdeva et al., 2021) estimates each sample's importance by its validation-loss variance, while *EL2N scores* (Paul et al., 2021) track a model's difficulty in learning particular data points. These methods all fit into a "score-and-sample" framework (Hastings, 1970), where the final selection depends on the magnitude of each item's quality score.

For a more detailed description see (Sachdeva et al., 2024). Below we describe the two algorithms proposed in (Sachdeva et al., 2024) and used as benchmarks in Section 5.

**ASK-LLM:** In ASK-LLM (Sachdeva et al., 2024), a proxy LLM is prompted with a potential training example and asked whether the example should be used for training. More specifically, the proxy LLM is provided the training example followed by the prompt "Does the previous paragraph contain informative signal for fine-tuning a large-language model? An informative datapoint should be well-formatted, contain some usable knowledge of the world, and strictly NOT have any harmful, racist, sexist, etc. content. OPTIONS: yes, no". It then takes the softmax probability of the token "yes" as the estimated data-quality score and sorts according to score to pick Top n data points.

**Density sampling:** (Sachdeva et al., 2024) assumes access to embeddings from a pre-trained LLM. Given a dataset $D$ it uses a kernel $k(x, y)$, to estimate the density using the following score.

$$\text{score}(y) = \sum_{x \in D} k_\lambda(x, y),$$

where $\lambda$ is a smoothing parameter and controls the scale of the data points' effects. Density Sampling then uses Inverse propensity sampling (IPS) to select items proportional to their re-weighted and normalized inverse score. The algorithm as provided in (Sachdeva et al., 2024) is summarized below.

**Clustering Based Sensitivity Sampling:** (Axiotis et al., 2024) The method uses k-means clustering and sensitivity sampling using the embedding representation of the data with respect to which the model loss is measured and ensures that the sampled elements' average loss corresponds to the average loss of the whole dataset. The algorithm as presented in (Axiotis et al., 2024) is summarized below.

---

**Algorithm 4** Clustering Based Sensitivity Sampling $(\mathcal{D}, k, \varepsilon, \Lambda, C)$ (Axiotis et al., 2024)

---

1: **Input:** a dataset $\mathcal{D}$ partitioned into clusters $C = (C_1, \ldots, C_k)$ with centers $c_1, \ldots, c_k$ and a $k$-tuple of parameters $\Lambda_1, \ldots, \Lambda_k$.
2: **for** $e \in C_i$ **do**
3:      Define $\hat{\ell}(e) := \ell(c_i)$ and $v(e) := \|e - c_i\|^z$.
4: Let $s := \lceil \varepsilon^{-2}(2 + 2\varepsilon/3) \rceil$. For $e \in C_i$ define $p_e := \frac{\hat{\ell}(e) + \Lambda_i v(e)}{\sum_i \Lambda_i \Phi(C_i, \{c_i\}) + \sum_{x \in \mathcal{D}} \hat{\ell}(x)}$ and $w(e) = s^{-1} p_e^{-1}$.
5: Compute a sample $S$ of $s$ points, picked independently following the distribution $p_e$.
6: **Output:** the set $S$ with weights $w$.

---

## B. Gradient and Hessian of the Loss

**Proposition B.1.** *Consider the Loss function as defined in (4) and suppose assumption 4.1 holds. Then the gradient and Hessian of $\mathcal{L}_\mathcal{S}$ are respectively given by*

$$\nabla \mathcal{L}_\mathcal{S}(\Theta) = \frac{1}{n} \sum_{i \in \mathcal{S}} \sum_{j \in [M_i]} \text{vec}\left( \mathbf{x}_{i,j} \otimes \left( p(\,\cdot\,|\mathbf{x}_{i,j}; \Theta) - \mathbb{1}(y_{i,j}) \right) \right)$$

$$\nabla^2 \mathcal{L}_\mathcal{S}(\Theta) = \frac{1}{n} \sum_{i \in \mathcal{S}} \sum_{j \in [M_i]} \left( \mathbf{diag}(p(\,\cdot\,|\mathbf{x}_{i,j}; \Theta)) - p(\,\cdot\,|\mathbf{x}_{i,j}; \Theta) p(\,\cdot\,|\mathbf{x}_{i,j}; \Theta)^\top \right) \otimes \mathbf{x}_{i,j} \mathbf{x}_{i,j}^\top$$

*Proof.* Recall that the loss function is given by

$$\mathcal{L}_\mathcal{S}(\Theta) = -\frac{1}{n} \sum_{i \in \mathcal{S}} \sum_{j \in [M_i]} \sum_{\ell \in [L]} \log P(y_{i,j} = \ell | \mathbf{x}_{i,j}, \Theta) \delta(y_{i,j} = \ell)$$

$$= -\frac{1}{n} \sum_{i \in \mathcal{S}} \sum_{j \in [M_i]} \sum_{\ell \in [L]} \log \left( \frac{\exp\left( (\Theta^T \mathbf{x}_{i,j})_\ell \right)}{\sum_{\ell'=1}^{L} \exp\left( (\Theta^T \mathbf{x}_{i,j})_{\ell'} \right)} \right) \delta(y_{i,j} = \ell).$$

Now the loss can be re-written as

$$\mathcal{L}_\mathcal{S}(\Theta) = \frac{-1}{n} \sum_{i \in \mathcal{S}} \sum_{j \in [M_i]} \left[ \theta_{y_{i,j}}^T \mathbf{x}_{i,j} - \log \sum_{\ell=1}^{L} \exp(\theta_\ell^T \mathbf{x}_n) \right]$$

Now note that

$$\frac{\partial}{\partial \theta_\ell} \theta_{y_{i,j}}^T \mathbf{x}_{i,j} = \delta(y_{i,j} = \ell) \mathbf{x}_{i,j}$$

and that,

$$\frac{\partial}{\partial \theta_\ell} \log \sum_{\ell'=1}^{L} \exp(\theta_{\ell'}^T \mathbf{x}_{i,j}) = \frac{\sum_{\ell'=1}^{L} \exp(\theta_{\ell'}^T \mathbf{x}_{i,j}) \times \delta(y_{i,j} = \ell) \mathbf{x}_{i,j}}{\sum_{k=1}^{L} \exp(\theta_k^T \mathbf{x}_{i,j})}$$

$$= \sum_{\ell'=1}^{L} p(y_{i,j} = \ell' | \mathbf{x}_{i,j}; \Theta) \delta(y_{i,j} = \ell) \mathbf{x}_{i,j}$$

$$= p(y_{i,j} = \ell | \mathbf{x}_{i,j}; \Theta) \mathbf{x}_{i,j}$$

Combining both we get

$$\frac{\partial}{\partial\theta_\ell}\mathcal{L}_\mathcal{S}(\Theta) = \frac{-1}{n} \sum_{i\in\mathcal{S}} \sum_{j\in[M_i]} \Big(\delta(y_{i,j}=\ell) - p(y_{i,j}=\ell|\mathbf{x}_{i,j};\Theta)\Big)\mathbf{x}_{i,j}$$

Therefore the gradient of the loss $\mathcal{L}_\mathcal{S}(\Theta)$ with respect to $\Theta$ is given by

$$\nabla\mathcal{L}_\mathcal{S}(\Theta) = \frac{-1}{n} \sum_{i\in\mathcal{S}} \sum_{j\in[M_i]} \text{vec}\Big(\mathbf{x}_{i,j} \otimes \big(\mathbb{1}(y_{i,j}) - p(y_{i,j}=\ell|\mathbf{x}_{i,j};\Theta)\big)\Big) \tag{11}$$

where $\mathbb{1}(y_{i,j}) \in \mathbb{R}^L$ is a one-hot vector with the $y_{i,j}$-th entry as 1 and $\otimes$ is the Kronecker product.

Next we compute the Hessian. Note that

$$\frac{\partial^2}{\partial\theta_\ell\theta_{\ell'}}\mathcal{L}_\mathcal{S}(\Theta) = \frac{-1}{n}\frac{\partial}{\partial\theta_\ell} \sum_{i\in\mathcal{S}} \sum_{j\in[M_i]} \Big(\delta(y_{i,j}=\ell) - p(y_{i,j}=\ell|\mathbf{x}_{i,j};\Theta)\Big)\mathbf{x}_{i,j}$$

$$= \frac{1}{n} \sum_{i\in\mathcal{S}} \sum_{j\in[M_i]} \left(\frac{\partial}{\partial\theta_{\ell'}}p(y_{i,j}=\ell|\mathbf{x}_{i,j};\Theta)\right)\mathbf{x}_{i,j}^T$$

$$= \frac{1}{n} \sum_{i\in\mathcal{S}} \sum_{j\in[M_i]} p(y_{i,j}=\ell|\mathbf{x}_{i,j};\Theta)\Big(\delta(\ell=\ell') - p(y_{i,j}=\ell|\mathbf{x}_{i,j};\Theta)\Big)\mathbf{x}_{i,j}\mathbf{x}_{i,j}^\top$$

and therefore, the Hessian of the loss is given by

$$\nabla^2\mathcal{L}_\mathcal{S}(\Theta) = \frac{1}{n} \sum_{i\in\mathcal{S}} \sum_{j\in[M_i]} \Big(\mathbf{diag}(p(\cdot|\mathbf{x}_{i,j};\Theta)) - p(\cdot|\mathbf{x}_{i,j};\Theta)p(\cdot|\mathbf{x}_{i,j};\Theta)^\top\Big) \otimes \mathbf{x}_{i,j}\mathbf{x}_{i,j}^\top \tag{12}$$

Now note that $p(\cdot|\mathbf{x}_{i,j};\Theta) \geq e^{-2\alpha}$ where $\sup_{\ell,i,j}\big|\Theta_\ell^\top\mathbf{x}_{i,j}\big| \leq \alpha$.

Therefore

$$\nabla^2\mathcal{L}_\mathcal{S}(\Theta) = \frac{1}{n} \sum_{i\in\mathcal{S}} \sum_{j\in[M_i]} \Big(e^{-2\alpha}I_L - e^{-4\alpha}\mathbf{1}\mathbf{1}^\top\Big) \otimes \mathbf{x}_{i,j}\mathbf{x}_{i,j}^\top \tag{13}$$

Assume $\Big(e^{-2\alpha}I_L - e^{-4\alpha}\mathbf{1}\mathbf{1}^\top\Big) \succeq \gamma\mathbf{I_L}$ for some $\gamma > 0$. Then we have

$$\nabla^2\mathcal{L}_\mathcal{S}(\Theta) \succeq \frac{1}{n} \sum_{i\in\mathcal{S}} \sum_{j\in[M_i]} \gamma I_L \otimes \mathbf{x}_{i,j}\mathbf{x}_{i,j}^\top \tag{14}$$

$\square$

## C. Proof of Error Bound

### C.1. Proof of Lemma 4.4

**Lemma 4.4.** *Suppose that Assumption 4.2 holds. Let $\mathcal{S}$ be the set of $n$ sentences generated by* `FisherSFT` *and* (7) *be their design matrix. Let $M = \max_{i\in[N]} M_i$. Then*

$$\max_{i\in[N]} \sum_{j=1}^{M_i} \|\mathbf{x}_{i,j}\|_{\bar{\Sigma}_\mathcal{S}^{-1}}^2 \leq \frac{\sigma_0^{-2}\log\left(1 + \frac{\sigma_0^{-2}nM}{d}\right)\kappa dM}{\log(1+\sigma_0^{-2})}\frac{}{n}.$$

*Proof.* We derive an upper bound on $\|\mathbf{x}_{i,j}\|_{\bar{\Sigma}_n^{-1}}$, where $\mathbf{x}_{i,j} \in \mathbb{R}^d$ is a feature vector and $\bar{\Sigma}_n \in \mathbb{R}^{d\times d}$ is a design matrix obtained by greedy log-determinant maximization. Let $\mathcal{D} = \{\mathbf{x}_{i,j} : i \in [N], j \in [M_i]\}$ be a dataset of $N$ data points such

that $\|\mathbf{x}_{i,j}\|_2 \leq 1$. Let $I_t \in [N]$ be the index of the $t$-th chosen feature vector and $\mathcal{S}_t = \{I_\ell\}_{\ell=1}^t$ be the first $t$ chosen feature vectors. For simplicity we use $\bar{\Sigma}_{\mathcal{S}}$ and $\bar{\Sigma}_n$ interchangeably. Let

$$\bar{\Sigma}_t = \sigma_0^2 \boldsymbol{I} + \sum_{i \in \mathcal{S}_t} \sum_{j=1}^{M_i} \mathbf{x}_{i,j} \mathbf{x}_{i,j}^\top$$

where $\sigma_0 > 0$ is a constant that guarantees that $\Sigma_0$ is well defined.

The $t$-th feature vector is chosen as

$$I_t = \operatorname*{argmax}_{i \in [N] \setminus \mathcal{S}_{t-1}} \log \det \left( \bar{\Sigma}_{t-1} + \sum_{j=1}^{M_t} \mathbf{x}_{t,j} \mathbf{x}_{t,j}^\top \right). \tag{15}$$

**Lemma C.1.** *For any $i \in [N]$ and $t \in [n]$,*

$$\sum_{j=1}^{M_i} \mathbf{x}_{i,j}^\top \bar{\Sigma}_t^{-1} \mathbf{x}_{i,j} \leq \sum_{j=1}^{M_i} \mathbf{x}_{i,j}^\top \bar{\Sigma}_{t-1}^{-1} \mathbf{x}_{i,j}.$$

*Proof.* Define the matrix

$$X = \begin{bmatrix} \mathbf{x}_{i,1} & \mathbf{x}_{i,2} & \cdots & \mathbf{x}_{i,M_t} \end{bmatrix},$$

so that each $\mathbf{x}_{i,j}$ is a column of $X$. Then we can write

$$\sum_{i=1}^{M_t} \mathbf{x}_{i,j} \mathbf{x}_{i,j}^T = X X^T.$$

Hence we want to find the inverse of

$$\bar{\Sigma}_{t-1} + X X^T.$$

Using **Sherman–Morrison–Woodbury identity**, which states that for an invertible matrix $A$ and any matrices $U, C, V$ of compatible dimensions (with $C$ also invertible), one has

$$(A + U C V)^{-1} = A^{-1} - A^{-1} U (C^{-1} + V A^{-1} U)^{-1} V A^{-1}.$$

In our case, we set

$$A = \bar{\Sigma}_{t-1}, \quad U = X, \quad C = I_{M_t}, \quad V = X^T,$$

where $I_{M_t}$ is the $M_t \times M_t$ identity matrix. Then

$$A + U C V = \bar{\Sigma}_{t-1} + X I_{M_t} X^T = \bar{\Sigma}_{t-1} + X X^T.$$

By applying the identity, we get

$$(\bar{\Sigma}_{t-1} + X X^T)^{-1} = \bar{\Sigma}_{t-1}^{-1} - \bar{\Sigma}_{t-1}^{-1} X (I_{M_t} + X^T \bar{\Sigma}_{t-1}^{-1} X)^{-1} X^T \bar{\Sigma}_{t-1}^{-1}.$$

which implies

$$\bar{\Sigma}_t^{-1} \preceq \bar{\Sigma}_{t-1}^{-1}.$$

Therefore we get $v^\top \bar{\Sigma}_t^{-1} v \leq v^\top \bar{\Sigma}_{t-1}^{-1} v$ for any vector $v \in \mathbb{R}^d$ which concludes the proof. $\qquad\square$

Lemma C.1 implies that

$$\sum_{j=1}^{M_i} \mathbf{x}_{i,j}^\top \bar{\Sigma}_n^{-1} \mathbf{x}_{i,j} \leq \frac{1}{n} \sum_{t=1}^n \sum_{j=1}^{M_i} \mathbf{x}_{i,j}^\top \bar{\Sigma}_t^{-1} \mathbf{x}_{i,j}.$$

holds for any $i \in [N]$. This allows us to attribute the quality of the solution to individual greedy steps in (15).

If the scope of the maximization was $i \in [N]$, the inequality $\sum_{j=1}^{M_i} \mathbf{x}_{i,j}^\top \bar{\Sigma}_{t-1}^{-1} \mathbf{x}_{i,j} \leq \sum_{j=1}^{M_t} \mathbf{x}_{I_t,j}^\top \bar{\Sigma}_{t-1}^{-1} \mathbf{x}_{I_t,j}$ would hold for any $i \in [N]$. Since the scope is $i \in [N] \setminus \mathcal{S}_{t-1}$, we make Assumption 4.2.

We also use the following logarithmic transformation.

**Lemma C.2.** *For any* $i \in [N]$ *and* $t \in [n]$,

$$\sum_{j=1}^{M_i} \mathbf{x}_{i,j}^\top \bar{\Sigma}_{t-1}^{-1} \mathbf{x}_{i,j} \leq \frac{\sigma_0^{-2} \log(1 + \mathbf{x}_{i,j}^\top \bar{\Sigma}_{t-1}^{-1} \mathbf{x}_{i,j})}{\log(1 + \sigma_0^{-2})}.$$

*Proof.* We start with an upper bound on $\sum_{j=1}^{M_i} \mathbf{x}_{i,j}^\top \bar{\Sigma}_{t-1}^{-1} \mathbf{x}_{i,j}$. By Weyl's inequalities, we have

$$\lambda_1(\bar{\Sigma}_{t-1}^{-1}) = \lambda_d^{-1}(\bar{\Sigma}_{t-1}) \leq \lambda_d^{-1}(\sigma_0^2 I_d) = \sigma_0^{-2}.$$

Therefore, under the assumption that $\|\mathbf{x}_{i,j}\|_2 \leq 1$, we have $\sum_{j=1}^{M_i} \mathbf{x}_{i,j}^\top \bar{\Sigma}_{t-1}^{-1} \mathbf{x}_{i,j} \leq \sigma_0^{-2} M_i$. Now note that for any $x \in [0, u]$,

$$x = \frac{x}{\log(1+x)} \log(1+x) \leq \left( \max_{x \in [0,u]} \frac{x}{\log(1+x)} \right) \log(1+x) = \frac{u}{\log(1+u)} \log(1+x).$$

Finally, we set $x = \sum_{j=1}^{M_i} \mathbf{x}_{i,j}^\top \bar{\Sigma}_{t-1}^{-1} \mathbf{x}_{i,j}$ and $u = \sigma_0^{-2} M_i$, and get our claim. $\square$

**Assumption C.3.** There exists a constant $\kappa \geq 1$ such that

$$\log\det\left(I_d + \sum_{j=1}^{M_i} \bar{\Sigma}_{t-1}^{-1/2} \mathbf{x}_{i,j} x_{i,j}^\top \bar{\Sigma}_{t-1}^{-1/2}\right) \leq \kappa \log\det\left(I_d + \sum_{j=1}^{M_{I_t}} \bar{\Sigma}_{t-1}^{-1/2} \mathbf{x}_{I_t,j} \mathbf{x}_{I_t,j}^\top \bar{\Sigma}_{t-1}^{-1/2}\right)$$

holds for any $i \in \mathcal{S}_{t-1}$ and $t \in [n]$.

Now we apply Assumption C.3 and Lemma C.2, use the telescoping property of the sum, and $M = \max_{i \in [N]} M_i$ to get

$$\sum_{t=1}^{n}\sum_{j=1}^{M_i}\mathbf{x}_{i,j}^{\top}\bar{\Sigma}_{t-1}^{-1}\mathbf{x}_{i,j} \overset{(a)}{\leq} \sum_{t=1}^{n}\sum_{j=1}^{M_i}\frac{\sigma_0^{-2}}{\log(1+\sigma_0^{-2})}\log(1+\mathbf{x}_{i,j}^{\top}\bar{\Sigma}_{t-1}^{-1}\mathbf{x}_{i,j})$$

$$\overset{(b)}{\leq}\frac{\sigma_0^{-2}}{\log(1+\sigma_0^{-2})}\sum_{t=1}^{n}\sum_{j=1}^{M_i}\log\det(I_d+\bar{\Sigma}_{t-1}^{-1/2}\mathbf{x}_{i,j}\mathbf{x}_{i,j}^{\top}\bar{\Sigma}_{t-1}^{-1/2})$$

$$\overset{(c)}{\leq}\frac{\sigma_0^{-2}M_i}{\log(1+\sigma_0^{-2})}\sum_{t=1}^{n}\log\det(I_d+\frac{1}{M_i}\sum_{j=1}^{M_i}\bar{\Sigma}_{t-1}^{-1/2}\mathbf{x}_{i,j}\mathbf{x}_{i,j}^{\top}\bar{\Sigma}_{t-1}^{-1/2})$$

$$\leq\frac{\sigma_0^{-2}M}{\log(1+\sigma_0^{-2})}\sum_{t=1}^{n}\log\det(I_d+\sum_{j=1}^{M_i}\bar{\Sigma}_{t-1}^{-1/2}\mathbf{x}_{i,j}\mathbf{x}_{i,j}^{\top}\bar{\Sigma}_{t-1}^{-1/2})$$

$$\overset{(d)}{\leq}\frac{\sigma_0^{-2}M}{\log(1+\sigma_0^{-2})}\sum_{t=1}^{n}\kappa\log\det(I_d+\sum_{j=1}^{M_{I_t}}\bar{\Sigma}_{t-1}^{-1/2}\mathbf{x}_{I_t,j}\mathbf{x}_{I_t,j}^{\top}\bar{\Sigma}_{t-1}^{-1/2})$$

$$\overset{(e)}{=}\frac{\kappa\sigma_0^{-2}M}{\log(1+\sigma_0^{-2})}\sum_{t=1}^{n}\log\det(\bar{\Sigma}_{t-1}+\sum_{j=1}^{M_{I_t}}\mathbf{x}_{I_t,j}\mathbf{x}_{I_t,j}^{\top})-\log\det(\bar{\Sigma}_{t-1})$$

$$\overset{(f)}{=}\frac{\kappa\sigma_0^{-2}M}{\log(1+\sigma_0^{-2})}\sum_{t=1}^{n}\log\det(\bar{\Sigma}_t)-\log\det(\bar{\Sigma}_{t-1})$$

$$\overset{(g)}{=}\frac{\kappa\sigma_0^{-2}M}{\log(1+\sigma_0^{-2})}(\log\det(\bar{\Sigma}_n)-\log\det(\bar{\Sigma}_0))$$

$$=\frac{\kappa\sigma_0^{-2}M}{\log(1+\sigma_0^{-2})}(\log\det(\bar{\Sigma}_n)-d\log(\sigma_0^2))$$

Here $(a)$ follows by Lemma C.2, $(b)$ follows by $\log(1+\mathbf{x}^{\top}A\mathbf{x})\leq\log\det\left(I+A^{1/2}\mathbf{x}\mathbf{x}^{\top}A^{1/2}\right)$ and $(c)$ follows by Jensen's inequality. Further $(d)$ follows by Assumption 4.2, $(e)$ follows by $\log\det\left(I+A^{-1/2}BA^{-1/2}\right)=\log\det(A+B)-\log\det(A)$, $(f)$ follows by the definition of $\bar{\Sigma}_t$ and finally $(g)$ follows by telescoping.

Furthermore,

$$\log\det(\bar{\Sigma}_n)\leq d\log\left(\frac{1}{d}\operatorname{tr}\left(\bar{\Sigma}_n\right)\right)=d\log\left(1+\frac{1}{d}\sum_{t=1}^{n}\operatorname{tr}\left(\sum_{j=1}^{M_{I_t}}\mathbf{x}_{I_t,j}\mathbf{x}_{I_t,j}^{\top}\right)\right)$$

$$=d\log\left(\sigma_0^2 I_d+\frac{1}{d}\sum_{t=1}^{n}\sum_{j=1}^{M_{I_t}}\mathbf{x}_{I_t,j}^{\top}x_{I_t,j}\right)\leq d\log\left(\sigma_0^2+\frac{nM}{d}\right).$$

Finally, we combine all claims and get

$$\max_{i\in[N]}\sum_{j=1}^{M_i}\mathbf{x}_{i,j}^{\top}\bar{\Sigma}_n^{-1}\mathbf{x}_{i,j}\leq\frac{\kappa}{n}\frac{\sigma_0^{-2}M}{\log(1+\sigma_0^{-2})}(d\log\det(\frac{1}{d}\operatorname{tr}(\sum_{t=1}^{n}\sum_{j=1}^{M_{I_t}}\mathbf{x}_{i,j}\mathbf{x}_{i,j}^{\top}))-d\log(\sigma_0))\leq\frac{\sigma_0^{-2}\log\left(1+\frac{\sigma_0^{-2}nM}{d}\right)}{\log(1+\sigma_0^{-2})}\frac{\kappa d}{n}.$$

This concludes the proof.

$\square$

## C.2. Proof of Lemma 4.5

**Lemma 4.5.** *Suppose that Assumption 4.1 holds and $\hat{\Theta}$ is the MLE in (5) such that $\hat{\Theta} \in \mathcal{B}$. Then there exists some $\alpha < 1$ such that*

$$\mathcal{L}_{\mathcal{S}}(\hat{\Theta}) - \mathcal{L}_{\mathcal{S}}(\Theta^*) - \langle \nabla \mathcal{L}_{\mathcal{S}}(\Theta^*), \hat{\Theta} - \Theta^* \rangle$$
$$\geq \frac{e^{-2\alpha}}{L} \left( \sum_{\ell=1}^{L} \|\hat{\theta}_\ell - \theta_\ell^*\|_{\bar{\Sigma}_{\mathcal{S}}} \right)^2.$$

*Proof.* Using Taylor's expansion

$$\mathcal{L}_{\mathcal{S}}(\Theta^*) + \langle \nabla \mathcal{L}_{\mathcal{S}}(\Theta_*), \hat{\Theta} - \Theta^* \rangle + \langle \hat{\Theta} - \Theta^*, \nabla^2 \mathcal{L}_{\mathcal{S}}(\Theta), \hat{\Theta} - \Theta^* \rangle = \mathcal{L}_{\mathcal{S}}(\hat{\Theta})$$

The Hessian is given by

$$\nabla^2 \mathcal{L}_{\mathcal{S}}(\Theta) = \frac{1}{n} \sum_{i \in \mathcal{S}} \sum_{j=1}^{M_i} (\mathrm{diag}(\mathbf{p}_{i,j}) - \mathbf{p}_{i,j}\mathbf{p}_{i,j}^\top) \otimes (\mathbf{x}_{i,j}\mathbf{x}_{i,j}^\top)$$

where $\mathbf{p}_{i,j} = p(\,\cdot\,|\mathbf{x}_{i,j};\Theta)$. Now using Claim 1 from (Hajek et al., 2014) we have

$$e^{2\alpha}(\mathrm{diag}(\mathbf{p}_{i,j}) - \mathbf{p}_{i,j}\mathbf{p}_{i,j}^\top) \succeq \frac{1}{L}\mathbf{I}_L + \frac{1}{L^2}\mathbb{1}\mathbb{1}^\top$$

where $\alpha = \max_{i,j} |\theta_{*,y_{i,j}}^\top \mathbf{x}_{i,j}| \leq 1$. Therefore we have

$$\nabla^2 \mathcal{L}_{\mathcal{S}}(\Theta) = \frac{1}{n} \sum_{i \in \mathcal{S}} \sum_{j=1}^{M_i} (\mathrm{diag}(\mathbf{p}_{i,j}) - \mathbf{p}_{i,j}\mathbf{p}_{i,j}^\top) \otimes (\mathbf{x}_{i,j}\mathbf{x}_{i,j}^\top)$$
$$\succeq \frac{1}{n} \sum_{i \in \mathcal{S}} \sum_{j=1}^{M_i} \left( \frac{e^{-2\alpha}}{L}\mathbf{I}_{L \times L} - \frac{e^{-2\alpha}}{L^2}\mathbb{1}\mathbb{1}^\top \right) \otimes (\mathbf{x}_{i,j}\mathbf{x}_{i,j}^\top)$$

Now consider $\langle \hat{\Theta} - \Theta^*, \nabla^2 \mathcal{L}_{\mathcal{S}}(\Theta), \hat{\Theta} - \Theta^* \rangle$. By defining $\Delta\Theta := \hat{\Theta} - \Theta^*$, we can express this as follows:

$$\langle \hat{\Theta} - \Theta^*, \nabla^2 \mathcal{L}_{\mathcal{S}}(\Theta), \hat{\Theta} - \Theta^* \rangle = \frac{1}{n} \sum_{i \in \mathcal{S}} \sum_{j=1}^{M_i} \sum_{k,k'} \left( \sqrt{\mathrm{diag}(\mathbf{p}_{i,j})}\Delta\Theta_{\cdot,k} \right)^\top \left( \sqrt{\mathrm{diag}(\mathbf{p}_{i,j})}\Delta\Theta_{\cdot,k'} \right) (\mathbf{x}_{i,j}\mathbf{x}_{i,j}^\top)_{k,k'}$$
$$- \langle \Delta\Theta_{\cdot,k}^\top \mathbf{p}_{i,j}, \Delta\Theta_{\cdot,k'}^\top \mathbf{p}_{i,j} \rangle (\mathbf{x}_{i,j}\mathbf{x}_{i,j}^\top)_{k,k'}$$
$$= \frac{1}{n} \sum_{i \in \mathcal{S}} \sum_{j=1}^{M_i} \left( \mathrm{Tr}\left( \sqrt{\mathrm{diag}(\mathbf{p}_{i,j})}\Delta\Theta^\top (\mathbf{x}_{i,j}\mathbf{x}_{i,j}^\top) \sqrt{\mathrm{diag}(\mathbf{p}_{i,j})}\Delta\Theta \right) \right.$$
$$\left. - \mathrm{Tr}\left( \mathbf{p}_{i,j}^\top \Delta\Theta \, \mathbf{x}_{i,j}\mathbf{x}_{i,j}^\top \Delta\Theta \, \mathbf{p}_{i,j} \right) \right)$$
$$= \frac{1}{n} \sum_{i \in \mathcal{S}} \sum_{j=1}^{M_i} \mathrm{Tr}\left( \mathbf{x}_{i,j}^\top \Delta\Theta \left( \mathrm{diag}(\mathbf{p}_{i,j}) - \mathbf{p}_{i,j}\mathbf{p}_{i,j}^\top \right) \Delta\Theta^\top \mathbf{x}_{i,j} \right)$$
$$\geq \frac{1}{n} \sum_{i \in \mathcal{S}} \sum_{j=1}^{M_i} \mathrm{Tr}\left( \mathbf{x}_{i,j}^\top \Delta\Theta \left( \frac{e^{-2\alpha}}{L}\mathbf{I}_{L \times L} - \frac{e^{-2\alpha}}{L^2}\mathbb{1}\mathbb{1}^\top \right) \Delta\Theta^\top \mathbf{x}_{i,j} \right)$$

Now observe that $\Delta\Theta\mathbb{1} = 0$ follows from Assumption 4.1 and solution $\hat{\Theta}$. Therefore,

$$
\begin{aligned}
\langle \hat{\Theta} - \Theta^*, \nabla^2\mathcal{L}_{\mathcal{S}}(\Theta), \hat{\Theta} - \Theta^* \rangle &\geq \frac{e^{-2\alpha}}{nL}\sum_{i\in\mathcal{S}}\sum_{j=1}^{M_i}\mathrm{Tr}(\Delta\Theta^\top \mathbf{x}_{i,j}\mathbf{x}_{i,j}^\top \Delta\Theta) \\
&= \frac{e^{-2\alpha}}{L}\mathrm{Tr}(\Delta\Theta^\top \bar{\Sigma}_{\mathcal{S}}\Delta\Theta) \\
&= \frac{e^{-2\alpha}}{L}\mathrm{Tr}\Big(\Delta\Theta^\top \sqrt{\bar{\Sigma}_{\mathcal{S}}}\sqrt{\bar{\Sigma}_{\mathcal{S}}}\Delta\Theta\Big) \\
&= \frac{e^{-2\alpha}}{L}\|\bar{\Sigma}\Delta\Theta\|_F^2 \\
&\geq \frac{e^{-2\alpha}}{L^2}\Bigg(\sum_{\ell}\|\hat{\theta}_\ell - \theta_\ell^*\|_{\bar{\Sigma}_{\mathcal{S}}}\Bigg)^2
\end{aligned}
$$

$\square$

## C.3. Proof of Lemma 4.6

**Lemma 4.6.** *With probability $1 - \delta$, the gradient of the loss satisfies*

$$
\sup_{\ell\in[L]}\big\|\nabla_\ell\mathcal{L}_{\mathcal{S}}(\Theta^*)\big\|_{\bar{\Sigma}_{\mathcal{S}}^{-1}} \leq C\sqrt{d + \log(L/\delta)}\,,
$$

*where $C > 0$ is some global constant.*

*Proof.* First observe that $\|\nabla_\ell\mathcal{L}_{\mathcal{S}}(\Theta)\|^2_{\bar{\Sigma}_{\mathcal{S}}^{-1}} = n\|\nabla_\ell\mathcal{L}_{\mathcal{S}}(\Theta)\|^2_{\Sigma_{\mathcal{S}}^{-1}}$ where $\Sigma_{\mathcal{S}} = \frac{1}{n}\bar{\Sigma}_{\mathcal{S}}$. Next recall that the gradient is given by

$$
\nabla\mathcal{L}_{\mathcal{S}}(\Theta) = \frac{1}{n}\sum_{i\in\mathcal{S}}\sum_{j\in[M_i]}\mathrm{vec}\Big(\mathbf{x}_{i,j}\otimes\big(p(\,\cdot\,|\mathbf{x}_{i,j};\Theta) - \mathbb{1}(y_{i,j})\big)\Big).
$$

Therefore

$$
\nabla_\ell\mathcal{L}_{\mathcal{S}}(\Theta) = \frac{1}{n}\sum_{i\in\mathcal{S}}\sum_{j\in[M_i]}\mathbf{x}_{i,j}\big(p(y_{i,j} = \ell|\mathbf{x}_{i,j};\Theta) - \mathbb{I}(y_{i,j} = \ell)\big).
$$

Define $X \in \mathbb{R}^{nM_i \times d}$ as the matrix whose rows are $\mathbf{x}_{i,j}, i \in \mathcal{S}, j \in [M_i]$, and $V^\ell$ be the $nM_i$ dimensional vector whose entries are $p(y_{i,j} = \ell|\mathbf{x}_{i,j};\Theta) - \mathbb{I}(y_{i,j} = \ell)$, i.e.,

$$
V_{ij}^\ell = \frac{\exp(\Theta_\ell^\top \mathbf{x}_{i,j})}{\sum_{\ell'=1}^{L}\exp(\Theta_{\ell'}^\top \mathbf{x}_{i,j})} - \mathbb{I}(y_{i,j} = \ell).
$$

Note that $\mathbb{E}[V^\ell] = 0$ and $\big|V_{ij}^\ell\big| \leq 2$, which implies $V$ is 4 sub-Gaussian. Therefore

$$
\big\|\nabla_\ell\mathcal{L}_{\mathcal{S}}(\Theta)\big\|^2_{\Sigma_{\mathcal{S}}^{-1}} = \frac{1}{n^2}(V^\ell)^\top X\Sigma_{\mathcal{S}}X^\top V^\ell \leq \frac{1}{n}\|V^\ell\|_2^2
$$

Using Bernstein's inequality, with probability $1 - \delta$, for some constant $C > 0$

$$
\big\|\nabla_\ell\mathcal{L}_{\mathcal{S}}(\Theta)\big\|^2_{\Sigma_{\mathcal{S}}^{-1}} \leq C\,\frac{(d + \log(1/\delta))}{n}
$$

Taking a union bound over all $\ell \in [L]$ we have with probability $1 - \delta$, for some constant $C > 0$

$$
\sup_{\ell\in[L]}\big\|\nabla_\ell\mathcal{L}_{\mathcal{S}}(\Theta)\big\|^2_{\Sigma_{\mathcal{S}}^{-1}} \leq C\,\frac{(d + \log(L/\delta))}{n}
$$

which implies we have with probability $1 - \delta$, for some constant $C > 0$

$$
\sup_{\ell\in[L]}\big\|\nabla_\ell\mathcal{L}_{\mathcal{S}}(\Theta)\big\|_{\bar{\Sigma}_{\mathcal{S}}^{-1}} \leq C\,\sqrt{(d + \log(L/\delta))}
$$

$\square$

