# OpenReview forum: "FisherSFT: Data-Efficient Supervised Fine-Tuning of Language Models Using Information Gain"
_ICML.cc/2025/Conference — ICML 2025 poster_

### Official Review · Reviewer_yVhv · 2025-03-09

**Overall Recommendation:** 3

**Summary:**

This paper presents an active learning methods to select the most informative examples in the training samples within a fixed budget, with the idea of consider greedily optimise the fisher information over the Hessian matrix with respect to the LLMs. To solve the issue under high-dimensional and large sample size, an efficient method is proposed by linking the setting to a multinomial logistic regression model. The authors performs some empirical experiments to verify their proposed method.

**Claims And Evidence:**

The claims are in general well supported by evidence, although I am not able to check section 4 (error bound). For experiments, since the initial claim is on high-dimensional and long sequence, can I ask the authors to provide synthetic examples other than L =20 and d = 10? and also it will be beneficial to show if we scale up the length or dimension, the proposed method is consistent better? Although it will be nice to show the difference between the computational gain comparing between algorithm 3.1 and 3.2. Same applies to experiment 4.2. Also, it is good to see experiment 4.3 on real-world data but then the setup is not very clear to me, perhaps it is worthwhile to map with one of the synthetic/semi-synthetic setup (or have this as an additional setup)?

**Essential References Not Discussed:**

I can not comments on this since I am not an expert.

**Experimental Designs Or Analyses:**

Please refers to my comments earlier.

**Methods And Evaluation Criteria:**

The general methods seems to be reasonable with some comments regarding experiment setup (see **Claims and Evidence**). Also, using LLMs as judge directly is not preferred and not convincing enough without any comparing with human experts (but this is less of a problem in my opinion).

**Other Comments Or Suggestions:**

Please see my comments earlier.

## update after rebuttal

I would like to thank all the authors for the further clarifications. I am in favour of accepting this paper, and my score remains unchanged.

**Other Strengths And Weaknesses:**

N/A

**Questions For Authors:**

Q 1: Regarding experiments setup and evaluation, detailed in previous comments?

Q2: This method is essentially utilising the Fisher information which is the biggest variance among the training samples. However, this most informative may not equivalent to the best training examples for particular type of problem (if the training samples are not well selected or if the problem is ill-defined). Could you comments on this and consider adding this as a part of the discussion?

**Relation To Broader Scientific Literature:**

This finding is interesting with direct links to classical statistical learning theory concepts (e.g. Fisher information and Bayesian modelling for multivariate Gaussian distribution) and provide an interesting view on selecting best training examples in high-dimensional space.

**Theoretical Claims:**

I am not able to check the details, but the general logic seems to be reasonable.

---

> ### Author Rebuttal · Authors · 2025-04-01
>
> We wanted to thank the reviewer for positive evaluation and appreciating that we bring classic ideas from statistical learning theory to LLMs. Our rebuttal is below. We focus on major issues and will incorporate all comments of the reviewer in the next version of our paper. If you have additional concerns, please reach out to us to discuss them.
>
> ### **Other Values of $d$ and $L$ in Synthetic Experiments**
>
> We plot the error curves for the synthetic problem in Section 5.1, and various $d \in \\{5, 10, 20, 30, 40\\}$ and $L \in \\{20, 30, 40, 50\\}$, at [anonymized link](https://imgur.com/a/9WvjPYu). TokenOD consistently performs better. We also show the trends for $L$ ([anonymized link](https://imgur.com/a/CL19WtK)) and $d$ ([anonymized link](https://imgur.com/a/kNKjio7)) when fixing the number of samples at $1000$.
>
> ### **Computational Gains in Fast Implementation of TokenOD**
>
> We compare the computation times of the fast and slow implementations of TokenOD on the problem in Section 5.3 at [anonymized link](https://imgur.com/a/J3RaNuH). The number of sentences is $5000$ and we plot the computation time for various values of $n$. The fast implementation of TokenOD is about $4$ times faster.
>
> ### **More Details on Experimental Setup in Section 5.3**
>
> The experiment can be described in our notation as follows. $x_{i,j}$ is the transformer embedding at the $j$-th token of the $i$-th sentence in the text corpus and $y_{i, j}$ is the identity of that token. For each compared method, we fine-tune a GPT-2 model on sentences collected by that method. At test time, we prompt the fine-tuned model with a few words from the original text corpus (represented by tokens $y_{i, 1}, y_{i, 2}, \dots, y_{i, p}$) and the model completes the sentence by generating $y_{i, p + 1}, y_{i, p + 2}, \dots, y_{i, 1024}$. Finally, we compare the quality of the completed sentences generated by the various models using an LLM as a judge.
>
> ### **Fisher Information**
>
> The Fisher information matrix in TokenOD is an approximation derived in Lemma 3.1. This algebraic form, an outer product of feature vectors, is the true Fisher information matrix for least squares. How is it useful beyond least squares? Simply put, optimization of this matrix leads to choosing feature vectors that cover all directions in training data uniformly. Because of that, it is possible to bound a worst-case prediction error over all feature vectors and be robust.
>
> ### **LLM as a Judge**
>
> We improved the evaluation, including reporting biases. See the rebuttal for [Reviewer M5jD](https://openreview.net/forum?id=e02oLEbehE&noteId=OaYaGwaft7) for details.

---

### Official Review · Reviewer_M5jD · 2025-03-13

**Overall Recommendation:** 4

**Summary:**

The paper presents a statistical approach to enhancing supervised fine-tuning efficiency through strategic training example selection, which stands in contrast to conventional random sampling methods. By conceptualizing the example selection problem as an optimal design task that maximizes the Hessian of the LLM's log-likelihood, the authors establish a theoretical framework that prioritizes examples with greater information value. The technical innovation lies in their efficient approximation of the LLM at the last layer using multinomial logistic regression models, which enables the development of TokenOD—a greedy algorithm that exploits log determinant submodularity to select sentences containing jointly informative tokens.

Their rigorous theoretical analysis demonstrates that the prediction error decreases at rate O(dL/√n), while extensive empirical evaluation across synthetic data, word embeddings, and GPT-2 fine-tuning on Shakespeare corpus consistently shows TokenOD outperforming baseline methods including uniform sampling and density-based approaches. The practical significance of this contribution becomes evident in the marked improvement in sample efficiency, with TokenOD frequently achieving with 1,000 examples what baseline methods require 2,000 examples to accomplish, thereby making the fine-tuning process substantially more resource-efficient while maintaining or improving the quality of generated text as evaluated by larger language models.

**Claims And Evidence:**

The claims in the paper "Autoregressive Optimal Design for Language Models" are generally supported by convincing evidence, though there are areas where the evidence could be strengthened.

The core claim that TokenOD improves statistical efficiency for supervised fine-tuning is well-supported through both theoretical analysis (Section 4's error bound theorem) and empirical results across multiple experimental settings. The synthetic experiments in Section 5.1 and word embedding experiments in Section 5.2 provide quantitative evidence showing clear improvements in both maximum and mean prediction errors compared to multiple baselines. These results consistently demonstrate TokenOD's superiority across different sample sizes.

The GPT-2 fine-tuning experiments provide additional evidence through LLM-based evaluation, showing that text generated by models trained on TokenOD-selected examples is preferred 56.5-77% of the time over baseline methods. While this evaluation approach using Claude 3 Sonnet as a judge is reasonable, it does introduce some subjectivity that could be acknowledged as a limitation.

**Essential References Not Discussed:**

n/a

**Experimental Designs Or Analyses:**

- in using LLM-as-a-judge, it would be better practice to hide the names of the two methods and to consider the effects of both orders. I don't think this would have a major effect but should be done in a revision. "better" is also ambiguously defined to the LLM

**Methods And Evaluation Criteria:**

The methods and evaluation criteria proposed in "Autoregressive Optimal Design for Language Models" are generally appropriate and well-designed for addressing the problem of improving supervised fine-tuning efficiency in LLMs.

**Other Comments Or Suggestions:**

- The paper would benefit from including discussion of limitations.

**Other Strengths And Weaknesses:**

n/a

**Questions For Authors:**

-

**Relation To Broader Scientific Literature:**

-

**Theoretical Claims:**

Overall, the theoretical claims and proofs (Lemma 3.1, Theorem 4) are sound and follow established techniques from matrix analysis and statistical learning theory.

---

> ### Author Rebuttal · Authors · 2025-04-01
>
> We wanted to thank the reviewer for positive evaluation, recognizing our contributions, and bringing up the prompt bias issue to our attention. Our rebuttal is below. We focus on major issues and will incorporate all comments of the reviewer in the next version of our paper. If you have additional concerns, please reach out to us to discuss them.
>
> ### **LLM as a Judge**
>
> We completely reworked the LLM-as-a-judge evaluation. The new prompt is
>
> > You are a judge of Shakespeare text. \
> ><tag1>text1</tag1> \
> ><tag2>text2</tag2> \
> > Respond 2 if the text inside <tag2> is more fluent Shakespeare text than the text inside <tag1>. Respond 1 otherwise.
>
> The prompt got simplified, does not name the methods, and targets our perceived benefit (improved language). We use a state-of-the-art LLM GPT-4o to judge. The text generated by the compared methods is randomized: one randomly-chosen method replaces text1 and the other text2. We tested the LLM judge and it chooses the first position with probability 0.54, which is slightly higher than 0.5 for a position-unbiased judge.
>
> In addition to improving evaluation, we added a new baseline Ask-LLM (Sachdeva et al., 2024). See Appendix A of our paper for more details on this method.
>
> We report the win rates of TokenOD on the Shakespeare dataset, as a function of sample size $n$, below:
>
> | TokenOD versus  | 100  |  200 |  500 | 1000 | 2000 | 5000 |
> |-----------------|------|------|------|------|------|------|
> | Uniform         | 0.80 | 0.56 | 0.60 | 0.59 | 0.64 | 0.74 |
> | DensitySampling | 0.61 | 0.66 | 0.68 | 0.62 | 0.54 | 0.84 |
> | Ask-LLM         | 0.59 | 0.52 | 0.68 | 0.59 | 0.68 | 0.74 |
>
> We observe that TokenOD consistently outperforms all baselines.
>
> We also added a new experiment on the Sherlock dataset from [Sherlock Holmes Next Word Prediction Corpus](https://www.kaggle.com/datasets/muhammadbilalhaneef/sherlock-holmes-next-word-prediction-corpus). The evaluation protocol is the same as in the Shakespeare dataset, except that "Shakespeare" is replaced with "Sherlock". The win rates of TokenOD on the Sherlock dataset are:
>
> | TokenOD versus  | 100  |  200 |  500 | 1000 | 2000 | 5000 |
> |-----------------|------|------|------|------|------|------|
> | Uniform         | 0.84 | 0.81 | 0.75 | 0.64 | 0.65 | 0.93 |
> | DensitySampling | 0.68 | 0.75 | 0.69 | 0.74 | 0.65 | 0.88 |
> | Ask-LLM         | 0.74 | 0.58 | 0.65 | 0.60 | 0.61 | 0.89 |
>
> We observe again that TokenOD consistently outperforms all baselines.

---

> > ### Comment · Reviewer_M5jD · 2025-04-08
> >
> > I appreciate the detailed response, including the new assessment and increasing the sample size to 5000. I recommend acceptance of the paper. The authors should consider including a discussion of limitations of the approach.

---

> > > ### Author Response · Authors · 2025-04-08
> > >
> > > Thank you for responding and having confidence in our work! We will definitely acknowledge and discuss limitations of the LLM-as-a-judge evaluation.

---

### Official Review · Reviewer_5LdG · 2025-03-16

**Overall Recommendation:** 3

**Summary:**

The paper studies the problem of data selection for the training (in particular, fine-tuning) of autoregressive language model. The data selection regime is based on an efficient estimation of a lower bound of the empirical fisher matrix. This estimation is then combined with a greedy optimal design algorithm for selecting data point for training: In particular, for each data point, the algorithm first computes the sum of the outer product the data point's embedding across all token positions, then the algorithm picks the data point that increases the volume of the design matrix most, and finally include this data point into the design matrix.
Then the authors further propose an accelerated version of the algorithm that avoids enumeration over all data points in the dataset.

The author then performed the evaluation on GPT2: Compared with past approaches that use the embedding of the sentences, the proposed approach shows a smaller error for all sizes of training dataset considered.

**Claims And Evidence:**

The paper claims that the proposed approach is efficient and a principled approach for dataset selection, and indeed the proposed algorithm and the empirical results provided confirm the claims.

**Essential References Not Discussed:**

I am not familiar with the literature. However, considering that the model only needs the pre read-out layer embedding of the sentence, I don't think there are lots of methods along this line, since most of them would require e.g. gradient information or an external model.

**Experimental Designs Or Analyses:**

The experiment design and analyses looks sensible to me.

**Methods And Evaluation Criteria:**

The proposed method intuitively makes a lot of sense and the evaluation setting: Supervised fine-tuning is a very practical and common problem/setting.

However, the fact that the algorithm does not rely on any gradient information is a little bit confusing. I did not check the derivation in detail and I believe in the correctness, but intuitively, you can use a quantity that only relies on the last layer embedding vectors as proxy for the fisher matrix, which depends on the gradient, is suprising.

The model considered is a little bit too small scale if I have to be picky: It would be better if the authors can demonstrate that the method works for e.g. LoRA fine-tuning of a 1M/3M model.

**Other Comments Or Suggestions:**

In line 112, input of Algorithm 1, the x denotes the sentence embedding rather than the "Sentences" themselves right?

Eq.4 should correspond to empirical fisher rather than the fisher matrix? Since there is no expectation with respect to all y values.

**Other Strengths And Weaknesses:**

N/A

**Questions For Authors:**

Would the data selection algorithm be biased by the length of the training sample? In line 5 of algorithm 1, the summation would contribute more as M_i , i.e. the number of tokens, increases?

**Relation To Broader Scientific Literature:**

The problem studied is of great interest, e.g. supervised fine-tuning is used in lots of RLHF pipeline and in LLM adaptation. The proposed method can be beneficial for many LLM practioner.

**Theoretical Claims:**

The paper provides a validation for Algorithm. 2 and the proposed lower bound of the fisher matrix in Section 3. and theoretical guarantee for the prediction error under the proposed data selection regime in Section 4. I did not check the proof details.

However it would be nice if the author could provide an empirical demonstration of the 1 / sqrt(n) convergence rate in the experiments.

---

> ### Author Rebuttal · Authors · 2025-04-01
>
> We wanted to thank the reviewer for positive evaluation, and recognizing the practicality of our solution as well as the importance of the solved problem. Our rebuttal is below. We focus on major issues and will incorporate all comments of the reviewer in the next version of our paper. If you have additional concerns, please reach out to us to discuss them.
>
> ### **Active Learning Using the Last-Layer Embedding**
>
> Model gradients are often used in active learning and bandit exploration ([Deep Batch Active Learning by Diverse, Uncertain Gradient Lower Bounds](https://arxiv.org/pdf/1906.03671), [Neural Contextual Bandits with UCB-based Exploration](https://arxiv.org/pdf/1911.04462)). We particularly note that such methods are computationally intensive since the parameter space of modern neural networks is huge in comparison to the last-layer embedding. Our method uses the last-layer embedding as a featurizer and is similar in spirit to [Neural Contextual Bandits with Deep Representation and Shallow Exploration](https://arxiv.org/pdf/2012.01780). This approach is known to be robust and has a much lower empirical regret than other uncertainty representation techniques ([Deep Bayesian Bandits Showdown: An Empirical Comparison of Bayesian Deep Networks for Thompson Sampling](https://arxiv.org/abs/1802.09127)).
>
> ### **Empirical Convergence Rate of TokenOD**
>
> We plot the error rate of TokenOD on the synthetic problem in Section 5.1 at [anonymized link](https://imgur.com/a/4HWu2oP). Specifically, we take the logarithm of the error rate and plot it as a function of the logarithm of the sample size. We observe a slope of $-0.3$, which means that the error rate is $O(n^p)$. We believe that this is close enough to the expected $p = -0.5$, especially since other factors may have played a role at our sample sizes.
>
> ### **More Extensive Empirical Evaluation**
>
> Our computational resources are limited and therefore we did not experiment beyond GPT-2. To partially address your concern, we further expanded our evaluation as follows. In addition to improving the LLM-as-a-judge evaluation, which other reviewers asked for, we added a new baseline Ask-LLM (Sachdeva et al., 2024). See Appendix A of our paper for more details on this method.
>
> We report the win rates of TokenOD on the Shakespeare dataset, as a function of sample size $n$, below:
>
> | TokenOD versus  | 100  |  200 |  500 | 1000 | 2000 | 5000 |
> |-----------------|------|------|------|------|------|------|
> | Uniform         | 0.80 | 0.56 | 0.60 | 0.59 | 0.64 | 0.74 |
> | DensitySampling | 0.61 | 0.66 | 0.68 | 0.62 | 0.54 | 0.84 |
> | Ask-LLM         | 0.59 | 0.52 | 0.68 | 0.59 | 0.68 | 0.74 |
>
> We observe that TokenOD consistently outperforms all baselines.
>
> We also added a new experiment on the Sherlock dataset from [Sherlock Holmes Next Word Prediction Corpus](https://www.kaggle.com/datasets/muhammadbilalhaneef/sherlock-holmes-next-word-prediction-corpus). The evaluation protocol is the same as in the Shakespeare dataset, except that "Shakespeare" is replaced with "Sherlock". The win rates of TokenOD on the Sherlock dataset are:
>
> | TokenOD versus  | 100  |  200 |  500 | 1000 | 2000 | 5000 |
> |-----------------|------|------|------|------|------|------|
> | Uniform         | 0.84 | 0.81 | 0.75 | 0.64 | 0.65 | 0.93 |
> | DensitySampling | 0.68 | 0.75 | 0.69 | 0.74 | 0.65 | 0.88 |
> | Ask-LLM         | 0.74 | 0.58 | 0.65 | 0.60 | 0.61 | 0.89 |
>
> We observe again that TokenOD consistently outperforms all baselines.
>
> ### **Bias Towards Longer Sentences**
>
> TokenOD is biased towards selecting longer sentences. This bias naturally arises because each token in a sentence is a training point in supervised fine-tuning, for the conditional probability of token $y_{i, j}$ given the history embedding $x_{i, j}$. In a sense, longer sentences can contribute more to information gain. However, the selection of the sentence depends on its total information gain in comparison to other available sentences. We also observe a similar bias towards longer sentences in the [GPT-2 code on Hugging Face](https://huggingface.co/transformers/v4.3.3/_modules/transformers/models/gpt2/modeling_gpt2.html#:~:text=loss_fct%20%3D%20CrossEntropyLoss,1)

---

> > ### Comment · Reviewer_5LdG · 2025-04-02
> >
> > I would like to thank the authors for the detailed response!
> >
> > I do not have further questions!

---

### Official Review · Reviewer_v3ni · 2025-03-25

**Overall Recommendation:** 3

**Summary:**

This paper casts the problem of data pruning for fine-tuning an LLM via SFT as one of optimal experiment design. In optimal experiment design, one wants to "probe" a system in a way that the combination of the probes you use are most effective in allowing you to extract the designed information from the system. By recognizing that the structure of modern LLMs has a penultimate layer that feeds into a linear layer that generates token logits, they treat the outputs of that layer as the relevant features for an input token. At that point, they invoke the idea of the Fisher information matrix of the loss to quantify quality. But this is intractable, and so by making a leap-of-faith technical assumption, they are able to get a lower bound on it by looking at something much much simpler --- just the sum of outer-products of the feature embeddings. Since maximizing log-det for such a sum has very nice submodularity properties, they are able to turn this into a pretty efficient greedy algorithm to select the "most informative" pieces of text to fine-tune on.

**Claims And Evidence:**

The fact that these simplifying assumptions lead to an efficient algorithm is undisputed. However, when one steps back, the radical boldness of what they are claiming is shocking and the evidence for that is weaker. Why? Because their method seems to never take into account in any way whether the LLM is already any good at producing the relevant sentences. Extremely low perplexity sentences and very high perplexity sentences are overtly treated the same!!!

But the proof of the pudding is in the eating, and they do experiments. However, there is a question I have regarding the use of LLM-as-judge here.

**Essential References Not Discussed:**

I was surprised to see the following paper not cited:

https://proceedings.neurips.cc/paper_files/paper/2022/hash/7b75da9b61eda40fa35453ee5d077df6-Abstract-Conference.html
Beyond neural scaling laws: beating power law scaling via data pruning

This takes a nuanced perspective on easy vs hard examples.

There is a sense in which the proposed approach in this paper is fundamentally about a different axis: picking a collectively loud set of examples vis-a-vis LLM feature space.

**Experimental Designs Or Analyses:**

Yes, see above.

**Methods And Evaluation Criteria:**

At one level, this all feels very reasonable. They're hand-picked example is striking, but they didn't tell us how that was picked so we can't trust that as an evaluation.

But there is a bigger question: their LLM-as-judge prompt literally labels one of the methods "Optimal Design" and asks the LLM to compare it against the other. Unless they repeated this evaluation with the labels flipped to check consistency, this violates one of the most basic principles of using any kind of evaluation --- you never prejudge the streams by giving them names that have any subjective valence or positivity/negativity. It's like asking a person whether they prefer the described actions of "Hero" vs "Villain."

**Other Comments Or Suggestions:**

None.

**Other Strengths And Weaknesses:**

I understand that one can always ask for more examples, but I am very surprised that the authors didn't anticipate the natural question: what if you use an LLM other than the one being fine-tuned to do the design? This transferability might be key in many practical scenarios where one might be using a black-box-API for doing the fine-tuning for a very strong model but have access to a weaker open-weights model to help with the choice of examples.

**Questions For Authors:**

Please report the results of what happens in the LLM-as-judge comparisons when you flip the labels? As well as when you call the two things "method A" and "method B"...

**Relation To Broader Scientific Literature:**

This is pretty decent

**Theoretical Claims:**

No.

---

> ### Author Rebuttal · Authors · 2025-04-01
>
> We wanted to thank the reviewer for positive evaluation and clearly summarizing the main technical contributions of our work. Our rebuttal is below. We focus on major issues and will incorporate all comments of the reviewer in the next version of our paper. If you have additional concerns, please reach out to us to discuss them.
>
> ### **LLM as a Judge**
>
> We completely reworked the LLM-as-a-judge evaluation. The new prompt is
>
> > You are a judge of Shakespeare text. \
> ><tag1>text1</tag1> \
> ><tag2>text2</tag2> \
> > Respond 2 if the text inside <tag2> is more fluent Shakespeare text than the text inside <tag1>. Respond 1 otherwise.
>
> The prompt got simplified, does not name the methods, and targets our perceived benefit (improved language). We use a state-of-the-art LLM GPT-4o to judge. The text generated by the compared methods is randomized: one randomly-chosen method replaces text1 and the other text2. We tested the LLM judge and it chooses the first position with probability 0.54, which is slightly higher than 0.5 for a position-unbiased judge.
>
> In addition to improving evaluation, we added a new baseline Ask-LLM (Sachdeva et al., 2024). See Appendix A of our paper for more details on this method.
>
> We report the win rates of TokenOD on the Shakespeare dataset, as a function of sample size $n$, below:
>
> | TokenOD versus  | 100  |  200 |  500 | 1000 | 2000 | 5000 |
> |-----------------|------|------|------|------|------|------|
> | Uniform         | 0.80 | 0.56 | 0.60 | 0.59 | 0.64 | 0.74 |
> | DensitySampling | 0.61 | 0.66 | 0.68 | 0.62 | 0.54 | 0.84 |
> | Ask-LLM         | 0.59 | 0.52 | 0.68 | 0.59 | 0.68 | 0.74 |
>
> We observe that TokenOD consistently outperforms all baselines.
>
> We also added a new experiment on the Sherlock dataset from [Sherlock Holmes Next Word Prediction Corpus](https://www.kaggle.com/datasets/muhammadbilalhaneef/sherlock-holmes-next-word-prediction-corpus). The evaluation protocol is the same as in the Shakespeare dataset, except that "Shakespeare" is replaced with "Sherlock". The win rates of TokenOD on the Sherlock dataset are:
>
> | TokenOD versus  | 100  |  200 |  500 | 1000 | 2000 | 5000 |
> |-----------------|------|------|------|------|------|------|
> | Uniform         | 0.84 | 0.81 | 0.75 | 0.64 | 0.65 | 0.93 |
> | DensitySampling | 0.68 | 0.75 | 0.69 | 0.74 | 0.65 | 0.88 |
> | Ask-LLM         | 0.74 | 0.58 | 0.65 | 0.60 | 0.61 | 0.89 |
>
> We observe again that TokenOD consistently outperforms all baselines.
>
> ### **How Is the Fine-Tuned LLM Used in Data Selection**
>
> Our data selection procedure depends on the LLM through embeddings $x_{i, j}$. They arise in both the log-likelihood in (2) and algorithm TokenOD. Simply put, TokenOD selects diverse sentences, which cover the embeddings $x_{i, j}$ more uniformly. As you pointed out, we reduce the original $d L \times d L$ Fisher information matrix into a $d \times d$ matrix, which can be optimized efficiently. While this neglects token-level prediction accuracy, it incorporates the LLM through the embeddings. A more direct optimization of the original matrix is an interesting direction that would require addressing the computational challenge.
>
> ### **Different Weaker LLM for Optimal Design**
>
> We agree that this is feasible when the embeddings of the weaker and stronger LLMs can be related. We have not done this in our work because we wanted to start with a simpler problem, the same LLM is used for both the optimal design and fine-tuning. We will discuss this option in the paper.
>
> ### **Missing Reference**
>
> Thank you for the reference. We will include it in the next version of the paper.

---

### Decision · Program_Chairs · 2025-05-01

**Decision:**

Accept (poster)

**Comment:**

This paper presents a method for selecting the most informative training examples for fine-tuning language models, using an efficient approximation of the Fisher information matrix to optimize the selection process. The reviewers praised the paper for its well-written presentation and interesting method, which demonstrates improved sample efficiency in fine-tuning language models. The reviewers' main criticisms centered around the experimental setup, the need for more details on the method's limitations, and the potential bias of the LLM-as-a-judge evaluation, with some reviewers also questioning the scalability of the approach and the lack of comparison to other methods.

The authors responded by providing additional experimental results, clarifying the technical details of their method, and addressing the concerns about the evaluation protocol. The reviewers indicated that the authors' rebuttal adequately addressed most of the criticisms, particularly those related to the evaluation protocol and the comparison to other methods. Therefore, the reviewers unanimously agreed to accept the paper. We would still recommend that the authors take the reviewers' feedback into account when preparing the camera-ready version.